# HYPERION: Fine-Grained Hypersphere Alignment for Robust Federated Graph Learning

**Guancheng Wan**[1†], **Xiaoran Shang**[1†], **Yuxin Wu**[4†],
**Guibin Zhang**[2], **Jinhe Bi**[3], **Liangtao Zheng**[5] ,
**Xin Lin**[5], **Yue Liu**[2], **Yanbiao Ma**[4], **Wenke Huang**[1*], **Bo Du**[1]
[1]Wuhan University    [2]NUS    [3]Ludwig-Maximilians-Universität München
[4]Renmin University of China    [5]UCSD
`guanchengwan@whu.edu.cn`

## Abstract

Robust Federated Graph Learning (FGL) provides an effective decentralized framework for training Graph Neural Networks (GNNs) in noisy-label environments. However, the subtlety of noise during training presents formidable obstacles for developing robust FGL systems. Previous robust FL approaches neither adequately constrain edge-mediated error propagation nor account for intra-class topological differences. At the client level, we innovatively demonstrate that hyperspherical embedding can effectively capture graph structures in a fine-grained manner. Correspondingly, our method effectively addresses the aforementioned issues through fine-grained hypersphere alignment. Moreover, we uncover undetected noise arising from localized perspective constraints and propose the geometric-aware hyperspherical purification module at the server level. Combining both level strategies, we present our robust FGL framework, `HYPERION`, which operates all components within a unified hyperspherical space. `HYPERION` demonstrates remarkable robustness across multiple datasets, for instance, achieving a $29.7\% \uparrow$ F1-macro score with $50\%$-pair noise on Cora. The code is available at: `https://github.com/GuanchengWan/HYPERION`.

## 1 Introduction

Federated Learning (FL) [20, 34] has recently emerged as a key area in decentralized machine learning. FL enables multiple clients to collaboratively train a shared global model while preserving data privacy [65, 20]. To leverage graph-structured data from diverse participants, Graph Neural Networks (GNNs) [22, 13, 31] have been integrated into FL, giving rise to Federated Graph Learning (FGL) [59, 49, 16, 45]. FGL combines two paradigms, effectively ensuring privacy[17, 19, 50] while enabling efficient distributed graph learning through neural message-passing mechanisms, which propagate node features and hidden representations in graph data.

As shown in Figure 1, although FGL offers numerous benefits [12, 2], it also introduces new vulnerabilities. Prior studies demonstrate that even minor structural or semantic perturbations can lead to misclassification in GNNs [7, 57, 68, 72]. These subtle differences may obscure critical information that defines node relationships and class boundaries. In FGL, coarse-grained representations of nodes within the same class can forcibly smooth out local topological differences, impeding the effective filtering of subtle noise and hindering the accurate capture of real semantic information and the underlying graph structure. Hence, we pose the following question: **I)** ***How can we learn class representations that are robust to noise while capturing subtle structural differences between***

---

† Equal Contribution.
* Corresponding Author.

39th Conference on Neural Information Processing Systems (NeurIPS 2025).

*similar nodes?* Such noise is not only inherently difficult to detect but also pervasive in graph data. Studies show that existing datasets can easily contain over 30% label errors [23, 41]. Recent FL methods address label noise via label correction [47, 60] and self-supervised learning [10, 58, 8], but these methods do not explicitly model the complex topological characteristics of graph data. Therefore, when dealing with graph data exhibiting complex topological structures, these approaches typically aggregate neighbor features indiscriminately, mixing noise with valid signals and degrading both alignment level and generalization performance. This leads to the question: **II)** *How can we adaptively identify and select high-confidence, stable nodes in each client's noisy graph data?*

Nevertheless, it is extremely difficult to completely remove noisy nodes solely relying on the client. The federated framework's privacy constraints limit each client's view to its local subgraph, preventing a global perspective. As a result, client models learn only local semantics and limited topological context, causing certain abnormal nodes to appear "normal" locally and evade detection. The limitation of this local perspective indirectly damages the generalization ability of the global model. Therefore, we ask: **III)** *How can we robustly refine semantic and topological knowledge during global aggregation and transfer it efficiently to the global model?*

To holistically address these challenges, we propose `HYPERION`: a **Hyperspherical-Embedding-Centric** Framework for Robust Federated Graph Learning, where all components operate on a unified hyperspherical space. To address issue **I)**, we introduce **Topological Prototypes Hyperspherical Learning (TP-HSL)** to fully capture the rich topological differences between nodes of the same class. Our method projects training node samples onto a hyperspherical embedding space. On the one hand, it maximizes the minimum spherical angle between different class prototype clusters, actively amplifying inter-class differences and enhancing the discriminability of decision boundaries. On the other hand, it minimizes the average spherical angle between nodes of the same class and their prototype centers to ensure tight intra-class clustering and strengthen structural correlations. Compared to conventional one-class-one-prototype approaches [46, 18, 51], TP-HSL provides finer structural

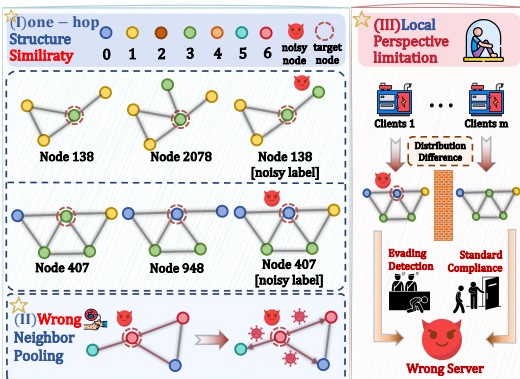

Figure 1: **Problem Illustration**. We describe the challenges FGL encounters under noisy labels: **I)** The coarse-grained representation method leads to ineffective differentiation between similar nodes, resulting in the coupling of noise and valid information. **II)** The edges between nodes in a graph facilitate the propagation of noise. **III)** The restricted view of individual clients leads to missed detection of certain noise.

modeling capabilities and heightened topological sensitivity. To address problem **II)**, we propose **Hyperspherical Consistency Noise Calibration (HS-CNC)**, which constructs a perturbed view of the graph. We retain only high-confidence nodes that consistently map to the same prototype cluster across views and filter out potential noisy nodes with unstable mappings. This process explicitly exposes the potential perturbation-sensitive areas through the "amplification of noise shifts" effect, ensuring information purity while effectively strengthening the correlation between nodes and the true topological structure. To solve issue **III)**, we propose **Geometric-Aware Hyperspherical Purification (GA-HSP)**. Driven by the Wasserstein distance, we distill, refine, and aggregate local prototype knowledge into robust global prototypes, constructing a well-defined global hypersphere. Then, by leveraging the covariance structure between the client hyperspheres and the global hypersphere, we apply the Mahalanobis distance to assess each node's outlier risk and eliminate drifting nodes biased by noise. To summarize, we make the following key contributions:

❶ *Problem Identification.* We study a challenging problem: overcoming the label noise in FGL. Our focus is on mitigating the negative influence of the label noise while overcoming several key limitations of existing solutions, including reliance on coarse-grained representations, neglect of the graph data's topological structure, and the absence of re-correction from a global perspective.

❷ *Practical Solution.* We introduce `HYPERION`, a novel and effective methodology that disentangles complex topological structures and mitigates malicious noise in FGL through hyperspherical representation. With the help of several technical innovations, `HYPERION` significantly enhances the model's ability to distinguish subtle structural differences while maintaining strong robustness.

❸ *Experimental Validation.* We conducted extensive experiments on five mainstream datasets under different noise types and ratios. The results demonstrate that our approach outperforms the state-of-the-art methods in multiple FGL environments. For instance, under the 50%-pair noise setting on the Cora dataset, our method achieves an impressive F1-macro score of 51.15%, outperforming the second-best method of 39.41% by a significant margin.

## 2 Preliminaries

**Notations.** Following the typical FGL framework, $M$ participants (indexed by m) collaboratively train a shared global model using their private graph data. Participant $m$ holds a graph $\mathcal{G}_m = (\mathcal{V}_m, \mathcal{A}_m, \mathcal{X}_m)$, where $\mathcal{V}_m = \{v_i\}_{i=1}^{N_m}$ is the node set containing $|\mathcal{V}_m| = N_m$ nodes, $\mathcal{A}_m \in \{0,1\}^{N_m \times N_m}$ is the adjacency matrix with $A_{ij} = 1$ if there is an edge between nodes $v_i$ and $v_j$ (and 0 otherwise), and $\mathcal{X}_m = \{x_i\}_{i=1}^{N_m}, x_i \in \mathbb{R}^d$ is the node feature set of dimension $d$. Moreover, $\mathcal{Y}_m \in \{0,1\}^{N_m \times C}$ is the label matrix, where each label $y_i \in \{0,1\}^C$ is a one-hot vector over $C$ classes. See Appendix A for detailed notation.

**Problem Formulation.** We focus on the semi-supervised node classification problem. Only a small set of nodes $\mathcal{V}_m^L$ is labeled for training, denoted as $\mathcal{V}_m^L = \mathcal{V}_m \setminus \mathcal{V}_m^U$, where $N_m^L$ is the number of labeled nodes. The remaining nodes are unlabeled and denoted as $\mathcal{V}_m^U$. Given $\mathcal{X}_m$ and $\mathcal{A}_m$, the goal of node classification is to train a classifier $f_{\theta_m} : (\mathcal{X}_m, \mathcal{A}_m) \to \hat{\mathcal{Y}}_m^L$, where the model parameters are optimized by minimizing the following objective:

$$\min_{\theta_m} \mathcal{L}(f_{\theta_m}(\mathcal{X}_m, \mathcal{A}_m), \mathcal{Y}_m^L), \tag{1}$$

where $\mathcal{L}$ is a loss function that measures the discrepancy between predictions and ground-truth labels. In this way, according to the Empirical Risk Minimization (ERM) principle, the well-trained classifier $f_{\theta_m}$ can generalize effectively to unseen nodes $\mathcal{V}_m^U$.

However, in real-world scenarios, the available labels $\mathcal{Y}_m^L$ may be corrupted, which degrades the generalization ability of the $m$-th client's classifier $f_{\theta_m}$. We denote these noisy labels as $\mathcal{Y}_m^N = \{\tilde{y}_1, \ldots, \tilde{y}_l\}$, where $\mathcal{Y}_m^L$ represents their corresponding ground-truth labels. To realistically model label noise in multi-source data, we consider two common types of label noise, defined as follows:

**Uniform noise [44]**: This noise model assumes that the true label has a probability $\in (0,1)$ of being uniformly flipped to any of the other classes with equal probability. Formally, for all $j \neq i$,

$$p(y_m^N = j \mid y_m^L = i) = \frac{\epsilon}{d-1}. \tag{2}$$

**Pair noise [63]**: This noise model assumes that the true label can only be flipped to a specific paired class with a fixed probability $\epsilon$, while remaining unchanged with probability $1 - \epsilon$.

The optimization objective is to learn a generalizable global model through the federated learning process that performs well under noisy conditions while maintaining strong robustness.

## 3 Methodology

### 3.1 Framework Overview

Inspired by our observations in Sec. 1 that FGL is sensitive to label noise, we propose `HYPERION` to finely enhance the model's ability to capture subtle structural differences among similar nodes and thereby improve robustness to noise. `HYPERION` comprises three key components: **I)** on each client, we extract local graph knowledge in a hyperspherical embedding space, where multiple class-specific prototype clusters capture fine-grained structural patterns; **II)** we select nodes whose embeddings remain consistently stable relative to their prototype clusters under perturbed views to ensure reliability; **III)** after the aggregation, we employ Wasserstein-driven prototype distillation and Mahalanobis-guided node purification at the server to refine and transfer complex structural knowledge. The detailed description of `HYPERION` is illustrated in Figure 2.

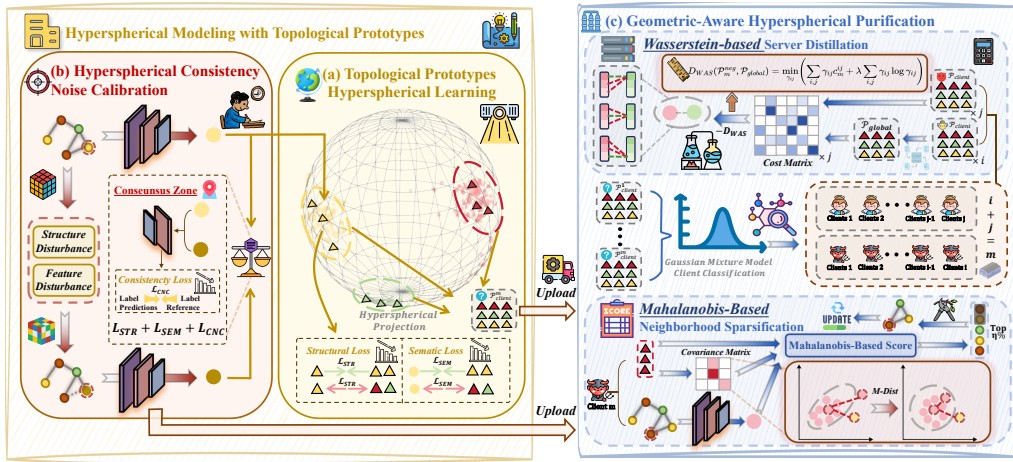

Figure 2: **Architecture illustration** of HYPERION. HYPERION comprises Topological Prototypes Hyperspherical Learning (TP-HSL), Hyperspherical Consistency Noise Calibration (HS-CNC) and Geometric-Aware Hyperspherical Purification (GA-HSP). Best viewed in color and zoom in for details.

## 3.2 Topological Prototypes Hyperspherical Learning (TP-HSL)

**Motivation.** Existing methods suffer from noise amplification due to their reliance on coarse Euclidean representations that fail to capture multiple structural variations among intra-class nodes. We address this limitation by introducing prototype clusters on the hypersphere, which reveal and mitigate noise propagation through fine-grained topological modeling.

**Hyperspherical Modeling with Topological Prototypes.** Each client projects its node features $\mathcal{X}_{m_i}$ and adjacency $\mathcal{A}_{m_i}$ into a unit hypersphere via an independent feature projector $\theta_m^p$:

$$z'_{m_i} = \theta_m^p(\mathcal{X}_{m_i}, \mathcal{A}_{m_i}), z_{m_i} = \frac{z'_{m_i}}{\|z'_{m_i}\|_2}. \tag{3}$$

The hyperspherical embedding representation $z_{m_i}$ can be modeled using the von Mises–Fisher (vMF) distribution [35, 55]. The vmF distribution is well-suited for accurately measuring the angular differences between embeddings, especially in high-dimensional spaces. In graph data, node embeddings are typically high-dimensional, and the vmF distribution, through its concentration parameter, effectively quantifies the similarity and dissimilarity between node embeddings [55, 21]. This geometric property makes the model more sensitive to meaningful semantic relationships while being less susceptible to noise and feature magnitude variations. Compared to traditional Euclidean spaces, the hyperspherical projection offers superior noise robustness by encoding semantic similarity in angular relationships rather than absolute positions, which is particularly crucial for handling label noise and structural variations in decentralized graph data. This approach allows the model to better capture fine-grained topological differences while maintaining strong generalization across clients.

However, prototype-based modeling with a single center suffers from inherently limited expressiveness [32], which fails to capture all the underlying topologies adequately. To solve the issue, we allocate a prototype cluster of shape $[M, C, K]$, defined as $\mathcal{P} = \{\mathbf{p}_m^{c,k} | c \in [C], k \in [K], m \in [M]\}$, which partitions the entire hyperspherical space into multiple topology-aware subspaces, each centered around a different prototype:

$$viewsp(z_{m_i}; \mathbf{w}_m^c, \mathcal{P}_m^c; \kappa) = \sum_{k=1}^{K} w_m^{c,k} Z_D(\kappa) \exp(\kappa \mathbf{p}_m^{c,k} \cdot z_{m_i}), \tag{4}$$

where $\mathbf{p}_m^{c,k}$ denotes the centroid prototype in the $k$-th topological subspace for class $c$ on the $m$-th client, $\omega_m^{c,k}$ denotes the corresponding prototype weight, and $\kappa$ is the concentration parameter. This approach ensures that the node embeddings within each class are better aligned with the corresponding prototypes, thereby improving both intra-class cohesion and inter-class distinction. By doing so, we can finely capture the structural differences within each class. For the input vector $z_i$, we further compute its prediction probability as follows:

$$p(y = y_i \mid z_{m_i}; \{\mathbf{w}_m^c, \mathcal{P}_m^c\}_{c=1}^C) = \frac{\sum_{k=1}^{K} w_m^{y_i,k} \exp(\kappa \mathbf{p}_m^{y_i,k} \cdot z_{m_i})}{\sum_{c=1}^{C} \sum_{k=1}^{K} w_m^{c,k} \exp(\kappa \mathbf{p}_m^{c,k} \cdot z_{m_i})}. \tag{5}$$

**Topological Prototypes Regularization Learning.** To strengthen inter-class separability and intra-class cohesion, we introduce several regularization terms based on the aforementioned class prediction probabilities. First, we maximize the minimal spherical angle among different class prototypes to optimize the distribution of inter-class prototypes further. Specifically, we calculate the cosine similarity matrices to measure the similarity between prototypes in Equation (6):

$$\mathbf{S}_m[i,j] = \exp\left(\frac{(\mathbf{p}_m^{c_i,k_i})^\top \mathbf{p}_m^{c_j,k_j}}{\|\mathbf{p}_m^{c_i,k_i}\|_2 \cdot \|\mathbf{p}_m^{c_j,k_j}\|_2}\right), \quad \forall i \in [1,C], j \in [1,K], \tag{6}$$

where $\mathbf{S}_m[i,j]$ quantifies the pairwise similarity between prototype $i$ and prototype $j$ within the hyperspherical space. As shown in Equation (7), we apply binary masks $\Gamma_m^{\text{pos}}$ and $\Gamma_m^{\text{neg}}$ to further categorize intra- and inter-class similarity:

$$\mathbf{S}_m^{\text{pos}} = \mathbf{S}_m \odot \Gamma_m^{\text{pos}}, \quad \mathbf{S}_m^{\text{neg}} = \mathbf{S}_m \odot \Gamma_m^{\text{neg}}, \tag{7}$$

where $\Gamma_m^{\text{pos}}$ activates entries corresponding to prototype pairs from the same class (excluding self-pairs), and $\Gamma_m^{\text{neg}}$ identifies prototypes pairs from different classes. The symbol $\odot$ represents the Hadamard product. Finally, we arrive at the following regularization term:

$$\mathcal{L}_{STR} = -\frac{1}{N}\sum_{i=1}^{N} \log\left(\frac{\sum_{j=1}^{N}\mathbf{S}_m^{\text{pos}}[i,j]}{\sum_{j=1}^{N}\mathbf{S}_m^{\text{pos}}[i,j] + \sum_{j=1}^{N}\mathbf{S}_m^{\text{neg}}[i,j] + \varepsilon}\right). \tag{8}$$

Here, $\varepsilon$ is a small smoothing factor used to prevent division by zero. This regularization term serves two key purposes: on the one hand, it boosts inter-class separability, thereby sharpening the decision boundaries; on the other hand, it ensures intra-class similarity, reinforcing the class semantic features.

To prevent global shifts in the semantic space formed by multiple prototypes, it is also essential to regulate the spherical angular relationship between embeddings and their associated prototypes. To this end, we encourage the minimization of the average spherical angle between node embeddings and their corresponding prototype cluster centers. This reinforces semantic consistency within each class. This regularization term can be modeled as:

$$\mathcal{L}_{\text{SEM}} = -\frac{1}{N}\sum_{i=1}^{N} log\frac{\sum_{k=1}^{K} w_m^{y_i,k}\exp(\kappa\mathbf{p}_m^{y_i,k}\cdot z_{m_i})}{\sum_{c=1}^{C}\sum_{k=1}^{K} w_m^{c,k}\exp(\kappa\mathbf{p}_m^{c,k}\cdot z_{m_i})}. \tag{9}$$

## 3.3 Hyperspherical Consistency Noise Calibration (HS-CNC)

**Motivation.** Due to the unique neighborhood diffusion mechanism in graph data, noisy labels tend to propagate along the edges. Therefore, it is crucial to design a noise node filtering mechanism that considers both structural and semantic aspects.

**Hyperspherical Robust Node Selection.** To effectively filter out potential noisy nodes whose mapping trajectories exhibit significant fluctuations, we assess the stability of nodes across different augmented graph views. Specifically, inspired by previous works [71, 27], we introduce data augmentation techniques: edge dropping and feature masking. These techniques randomly drop edges and certain feature values in the graph $\mathcal{G}_m(\mathcal{X}_m, \mathcal{A}_m)$:

$$\tilde{\mathcal{A}}_m = \mathcal{A}_m \odot \Gamma_m^{\mathcal{A}}, \tilde{\mathcal{X}}_m = \mathcal{X}_m \odot \Gamma_m^{\mathcal{X}}, \tag{10}$$

where $\Gamma_m^{\mathcal{A}} \in \{0,1\}^{N_m * N_m}$ is the randomly generated edge mask matrix, and $\Gamma_m^{\mathcal{X}} \in \{0,1\}^{N_m * d}$ is the randomly generated feature mask matrix.

By applying the two mask matrices, the augmented graph $\tilde{\mathcal{G}}_m(\tilde{\mathcal{X}}_m, \tilde{\mathcal{A}}_m)$ contains perturbed structural and semantic information. We assess the stability of each node by calculating the consistency of its representation across different views. Nodes that exhibit high consistency across both views are considered "clean nodes" because they maintain stable and consistent semantic representations under different perturbation conditions. We train using the subset of stable nodes $\mathcal{X}'_m$, which fundamentally ensures that learned representations are grounded in meaningful:

$$L_{\text{CNC}} = \sum_{v_m^i \in \mathcal{V}_m} \mathbb{1}\left(f_m(\mathcal{G}_m, v_m^i) = f_m(\tilde{\mathcal{G}}_m, v_m^i)\right) \cdot L\left(f_m(\mathcal{G}_m, v_m^i), \mathcal{Y}_m(v_m^i)\right), \tag{11}$$

where $\mathbb{1}$ is an indicator function that outputs 1 if the predicted results for node $v_m^i$ in both graphs are consistent, and 0 otherwise. The loss function $L$ computes the discrepancy between the node's predicted value and its true label and then calculates the gradient to update the parameters.

This design naturally connects to two theoretical perspectives. From the *information bottleneck* viewpoint, random edge dropping acts as compression: by pruning away connections, the model must retain relationships that consistently survive perturbations, which more likely reflect the task-relevant core structure [5, 15]. From the *generalization under noise* viewpoint, these augmentations inject structured noise that preserves global characteristics but prevents overfitting to fragile details. Consequently, the model is nudged toward flatter minima in the loss landscape, leading to improved robustness and generalization. Such principles align with findings in graph contrastive learning [62, 53]. As an example, `HYPERION` algorithm is shown in Algorithm 1.

---

**Algorithm 1** `HYPERION` Framework

---

Communication rounds $T$, participant scale $M$, $m$-th client private model $\theta_m$, $m$-th client local data $\mathcal{G}_m$, $m$-th client prototype cluster $\mathcal{P}_m$ and loss weight $\alpha, \beta$

The final global model $\theta_{global}$

**for** $t = 1, 2, \cdots, T$ **do**

    *Client Side:* **for** $m = 1$ ***to*** $M$ ***in parallel*** **do**

        $\mathcal{L}_{CNC} \leftarrow$ HypersphericalNoiseCalibration$(\mathcal{G}_m, \mathcal{P}_m)$by Equation (11)   // Select robust nodes and train with them

        $\mathbf{S}_m \leftarrow$ CalculateSimilarity$(\mathcal{P}_m)$by Equation (6)  // Calculate prototypes similarity metrix

        $\mathcal{L}_{STR} \leftarrow$ StructLoss$(\mathbf{S}_m)$by Equation (8)  // Calculate loss with inter- and intra-class prototypes

        $\mathcal{L}_{SEM} \leftarrow$ SemanticLoss$(\mathcal{P}_m)$by Equation (9)  // Calculate loss with embedding vector and prototypes

        $\theta_m^{t+1} \leftarrow$ LocalUpdating$(\theta_m^t, \mathcal{L}_{CNC} + \alpha\mathcal{L}_{STR} + \beta\mathcal{L}_{SEM})$  // Backward propagation

    *Server Side:*

    $\theta^{pos}, \theta^{neg} \leftarrow$ GMM$(\mathcal{P})$ // Client classification by prototypes

    $\mathcal{G}_m^{neg\prime} =$ NeighborhoodSparsification$(\theta_m^{neg}, \mathcal{G}_m), \forall m$  by Equation (18) // pruning with Mahalanobis distance

    $\theta_{global}, \mathcal{P}_{global} \leftarrow$ Aggregate$(\theta_m^{pos}, \mathcal{P}^{pos}), \forall m$  // Clean clients hyperspherical aggregation

    $\mathcal{P}'_{global} \leftarrow$ ServerDistillation$(\mathcal{P}_m^{neg}, \mathcal{P}_{global}), \forall m$ by Equation (13) // Wassertein distance Server Distillation

    $\theta_m \leftarrow \theta_{global}, \forall m$ // Distribute parameters to clients

**return** $\theta_{global}$

---

### 3.4 Geometric-Aware Hyperspherical Purification (GA-HSP)

**Motivation.** It is exceptionally challenging to entirely remove noisy nodes relying only on the client-side. Due to the non-IID distribution, each client has a limited perspecitve, meaning that some noisy nodes are detected as normal locally, but are considered anomalous when viewed globally. To address this issue, we propose GA-HSP, which performs knowledge purification on the server side.

**Prototype-based Client Classification.** Existing research suggests that, in practical scenarios, the data noise ratio at each client may vary to some extent [60]. To effectively identify noisy clients, we design an unsupervised detection method that fully exploits the global distribution characteristics of local prototypes on the client side, distinguishing between benign and malicious clients.

After training on each client, the local prototype clusters $\mathcal{P}$ are uploaded to the server, where the server computes the Gaussian Mixture Model (GMM) of the prototype clusters $\mathcal{P}$ from all $M$ clients.

$$p(\mathcal{P}) = \sum_{l=1}^{2} \pi_l \cdot \mathcal{N}(\mathcal{P} \mid \boldsymbol{\mu}_l, \boldsymbol{\Sigma}_l), \tag{12}$$

where $\pi_l$, $\boldsymbol{\mu}_l$ and $\boldsymbol{\Sigma}_l$ represent the weight, mean, and covariance matrix of the $l$-th Gaussian component, respectively. The client set $\phi$ is partitioned into two subsets: clean clients $\phi_{pos}$ and malicious clients $\phi_{neg}$. To ensure the reliability of the global knowledge, we apply GMM to re-aggregate the prototypes from the clean clients $\phi_{pos}$. For each class, we aim to aggregate the prototypes into $K$ clusters and take the center of each cluster as the global prototype for that class: $\mathcal{P}_{global} = \{p^{c,k} | c \in [C], k \in [K]\}$. Through this unsupervised approach, we can not only effectively filter out malicious clients, but also integrate the local information from all clean clients, thereby constructing a more precise global prototype cluster $\mathcal{P}_{global}$.

**Wasserstein-based Server Distillation.** To further filter out erroneous information in the prototypes, inspired by recent work on negative distillation [33], we adopt a "negative distillation" strategy: each malicious client prototype cluster $\mathcal{P}_m^{neg}$ serves as the teacher, while the global prototype cluster $\mathcal{P}_{global}$ functions as the student. The core of this distillation is to constrain the angular distance between the teacher and the student in the hyperspherical space, thereby suppressing and correcting anomalous representations in the global prototype cluster.

Traditional distillation methods fail to capture cross-dimensional similarities, leading to the underutilization of dimensional information. To address this issue, we propose a Wasserstein distance-driven prototype negative distillation method, which deploys the discrete Wasserstein distance to comprehensively measure the distributional differences between the teacher and student models. For the $m$-th teacher prototype cluster $\mathcal{P}_m^{neg}$, we define the discrete Wasserstein distance $D_{WAS}$ as follows:

$$D_{WAS}(\mathcal{P}_m^{neg}, \mathcal{P}_{global}) = \min_{\gamma_{ij}} \left( \sum_{i,j} \gamma_{ij} c_m^{ij} + \lambda \sum_{i,j} \gamma_{ij} \log \gamma_{ij} \right), \tag{13}$$

where $\gamma_{ij}$ represents the mass transferred from the teacher prototype cluster's dimension $q_i$ to the student prototype cluster's dimension $q_j$, subject to the constraints:

$$\sum_j \gamma_{ij} = \mathcal{P}_{m,i}^{neg}, \sum_i \gamma_{ij} = \mathcal{P}_{global,j}, \gamma_{ij} \geq 0, \tag{14}$$

where $\lambda$ is a hyperparameter controlling the entropy regularization term. A key component of this formulation is the cost matrix $c^m$, which encapsulates the dissimilarity between prototype dimensions:

$$c_m^{ij} = 1 - \frac{\mathcal{P}_{m,i}^{neg} \cdot \mathcal{P}_{global,j}}{\|\mathcal{P}_{m,i}^{neg}\| \|\mathcal{P}_{global,j}\|}. \tag{15}$$

The higher the similarity between prototype dimensions, the lower the transfer cost. Conversely, when there is a significant difference in the direction of the dimensions, the cost increases considerably. By minimizing $D_{WAS}$, the probability mass is effectively reallocated between proximate dimensions in the feature space, thereby naturally reinforcing the correlation of benign features between global prototypes, while effectively diminishing the impact of anomalous features.

**Mahalanobis-Based Neighborhood Sparsification.** Building on the negative distillation from malicious clients, we attempt to incorporate data pruning to purify them. Our out-of-distribution (OOD) detection method is based on Mahalanobis distance [32, 42], which is equivalent to the Euclidean distance scaled by the eigenvalues in the feature space. By introducing the inverse of the covariance matrix, the Mahalanobis distance can automatically adjust the importance of each dimension, avoiding certain dimensions dominating the distance calculation due to scale differences. In geometry, Mahalanobis distance transforms the data space into a standardized space, making distance calculations more equitable. We identify and remove noise nodes that deviate from the normal distribution by calculating the Mahalanobis distance between node embeddings and the global prototype distribution (as shown in Equation (18)). This method effectively filters out noise that is not detected by clients due to local perspective limitations. Inspired by the work [1], we simultaneously measure both inter-class and intra-class distances of prototype clusters to comprehensively assess the outlier risk of nodes. The inter-class prototype distance is defined as:

$$d_{\text{inter}}(z_{m_i}) = \frac{1}{|C|-1} \sum_{c \neq y_i} \left[ \frac{1}{\min_k \left( (z_{m_i} - p_m^{c,k})^\top \Lambda_c^{-1} (z_{m_i} - p_m^{c,k}) \right) + \varepsilon} \right], \tag{16}$$

Table 1: **Comparison with the state-of-the-art methods** on five selected real-world datasets. For each dataset, we report accuracy (%) and F1-macro (%) (with red/green markers indicating regression/improvement over FedAvg). The noise type is set to **50%-uniform** (upper) and **50%-pair** (lower). The best and second-best results are highlighted with **bold** and underline, respectively. Additional experimental results on more settings can be found in Appendix D.

| Category | Methods | Cora | | CiteSeer | | PubMed | | Physics | | Amazon_ratings | |
|---|---|---|---|---|---|---|---|---|---|---|---|
| | Metrics | ACC | F1-macro | ACC | F1-macro | ACC | F1-macro | ACC | F1-macro | ACC | F1-macro |
| FL | FedAvg [ASTAT17] | $32.91_{\uparrow00.00}$ | $30.47_{\uparrow00.00}$ | $28.15_{\uparrow00.00}$ | $26.98_{\uparrow00.00}$ | $59.39_{\uparrow00.00}$ | $57.19_{\uparrow00.00}$ | $70.69_{\uparrow00.00}$ | $39.21_{\uparrow00.00}$ | $33.65_{\uparrow00.00}$ | $\underline{25.07}_{\uparrow00.00}$ |
| | FedProx [MLSys20] | $34.64_{\uparrow01.73}$ | $33.12_{\uparrow02.65}$ | $26.67_{\downarrow01.48}$ | $25.84_{\downarrow01.14}$ | $61.31_{\uparrow01.92}$ | $58.77_{\uparrow01.58}$ | $\underline{72.53}_{\uparrow01.84}$ | $\mathbf{57.80}_{\uparrow18.59}$ | $35.86_{\uparrow02.21}$ | $24.94_{\downarrow00.13}$ |
| | FedNova [NeurIPS20] | $37.48_{\uparrow04.57}$ | $35.33_{\uparrow04.86}$ | $31.56_{\downarrow03.41}$ | $30.46_{\downarrow03.48}$ | $63.49_{\uparrow04.10}$ | $61.61_{\uparrow04.42}$ | $57.37_{\downarrow13.32}$ | $26.80_{\downarrow12.41}$ | $36.61_{\uparrow02.96}$ | $24.18_{\downarrow00.89}$ |
| | MOON [CVPR21] | $33.64_{\uparrow00.73}$ | $31.86_{\uparrow01.39}$ | $27.26_{\downarrow00.89}$ | $26.78_{\downarrow00.20}$ | $70.14_{\uparrow10.75}$ | $58.89_{\uparrow01.70}$ | $59.06_{\downarrow11.63}$ | $13.46_{\downarrow25.75}$ | $\mathbf{38.49}_{\uparrow04.84}$ | $20.67_{\downarrow04.40}$ |
| FGL | FGSSL [IJCAI23] | $36.65_{\uparrow03.74}$ | $34.43_{\uparrow03.96}$ | $32.74_{\uparrow04.59}$ | $32.21_{\uparrow05.23}$ | $65.92_{\uparrow06.53}$ | $71.59_{\uparrow14.40}$ | $50.76_{\downarrow19.93}$ | $13.78_{\downarrow25.43}$ | $38.43_{\uparrow04.78}$ | $20.49_{\downarrow04.58}$ |
| | FedGTA [VLDB24] | $31.54_{\downarrow01.37}$ | $29.84_{\downarrow00.63}$ | $28.44_{\uparrow00.29}$ | $27.40_{\uparrow00.42}$ | $57.41_{\downarrow01.98}$ | $56.63_{\downarrow00.56}$ | $60.48_{\downarrow10.21}$ | $25.93_{\downarrow13.28}$ | $33.71_{\uparrow00.06}$ | $24.46_{\downarrow00.61}$ |
| | FedTAD [IJCAI24] | $31.26_{\downarrow01.65}$ | $29.53_{\downarrow00.94}$ | $28.59_{\uparrow00.44}$ | $26.56_{\downarrow00.42}$ | $58.81_{\downarrow00.58}$ | $56.56_{\downarrow00.63}$ | $57.12_{\downarrow13.57}$ | $28.24_{\downarrow10.97}$ | $32.12_{\downarrow01.53}$ | $24.60_{\downarrow00.47}$ |
| Robust FL | FedNoRo [IJCAI23] | $32.27_{\downarrow00.64}$ | $30.31_{\downarrow00.16}$ | $28.44_{\uparrow00.29}$ | $27.09_{\uparrow00.11}$ | $60.25_{\uparrow00.86}$ | $56.85_{\downarrow00.34}$ | $70.74_{\uparrow00.05}$ | $39.59_{\uparrow00.38}$ | $33.78_{\uparrow00.13}$ | $24.16_{\downarrow00.91}$ |
| | FedNed [AAAI24] | $32.82_{\downarrow00.09}$ | $29.83_{\downarrow00.64}$ | $30.52_{\uparrow02.37}$ | $28.87_{\uparrow01.89}$ | $57.41_{\downarrow01.98}$ | $55.78_{\downarrow01.41}$ | $64.48_{\downarrow06.21}$ | $32.09_{\downarrow07.12}$ | $34.00_{\uparrow00.35}$ | $\mathbf{25.23}_{\uparrow00.16}$ |
| | FedCorr [CVPR22] | $34.37_{\uparrow01.46}$ | $28.44_{\downarrow02.03}$ | $22.20_{\downarrow05.95}$ | $24.08_{\downarrow02.90}$ | $57.36_{\downarrow02.03}$ | $56.12_{\downarrow00.69}$ | $60.73_{\downarrow09.96}$ | $25.12_{\downarrow14.09}$ | $35.22_{\uparrow01.57}$ | $24.02_{\downarrow01.05}$ |
| Robust GL | CRGNN [NN24] | $44.61_{\uparrow11.70}$ | $\underline{39.41}_{\uparrow08.94}$ | $39.55_{\uparrow11.40}$ | $36.21_{\uparrow09.23}$ | $\underline{73.66}_{\uparrow14.27}$ | $\underline{72.33}_{\uparrow15.14}$ | $55.45_{\downarrow15.24}$ | $22.73_{\downarrow16.48}$ | $36.41_{\uparrow02.76}$ | $10.68_{\downarrow14.39}$ |
| | RTGNN [WWW23] | $\underline{47.81}_{\uparrow14.90}$ | $38.72_{\uparrow08.25}$ | $\underline{41.93}_{\uparrow13.78}$ | $\underline{36.33}_{\uparrow09.35}$ | $67.33_{\uparrow07.94}$ | $44.09_{\downarrow13.10}$ | $66.29_{\downarrow04.40}$ | $29.05_{\downarrow10.16}$ | $36.65_{\uparrow03.00}$ | $21.52_{\downarrow03.55}$ |
| | CLNode [WSDM23] | $35.10_{\uparrow02.19}$ | $33.13_{\uparrow02.66}$ | $30.37_{\uparrow02.22}$ | $30.40_{\uparrow03.42}$ | $54.73_{\downarrow04.66}$ | $52.14_{\downarrow05.05}$ | $66.29_{\downarrow04.40}$ | $29.61_{\downarrow09.60}$ | $31.35_{\downarrow02.30}$ | $24.43_{\downarrow00.64}$ |
| Robust FGL | HYPERION | $\mathbf{53.56}_{\uparrow20.65}$ | $\mathbf{51.15}_{\uparrow20.68}$ | $\mathbf{47.11}_{\uparrow18.96}$ | $\mathbf{40.25}_{\uparrow13.27}$ | $\mathbf{74.85}_{\uparrow15.46}$ | $\mathbf{73.74}_{\uparrow16.55}$ | $\mathbf{75.21}_{\uparrow04.52}$ | $\underline{45.71}_{\uparrow06.50}$ | $38.96_{\uparrow05.31}$ | $22.77_{\downarrow02.30}$ |

| Category | Methods | Cora | | CiteSeer | | PubMed | | Physics | | Amazon_ratings | |
|---|---|---|---|---|---|---|---|---|---|---|---|
| | Metrics | ACC | F1-macro | ACC | F1-macro | ACC | F1-macro | ACC | F1-macro | ACC | F1-macro |
| FL | FedAvg [ASTAT17] | $33.27_{\uparrow00.00}$ | $31.97_{\uparrow00.00}$ | $30.67_{\uparrow00.00}$ | $30.23_{\uparrow00.00}$ | $49.75_{\uparrow00.00}$ | $49.39_{\uparrow00.00}$ | $49.93_{\uparrow00.00}$ | $33.81_{\uparrow00.00}$ | $34.94_{\uparrow00.00}$ | $21.69_{\uparrow00.00}$ |
| | FedProx [MLSys20] | $37.02_{\uparrow03.75}$ | $34.76_{\uparrow02.79}$ | $33.04_{\uparrow02.37}$ | $32.42_{\uparrow02.19}$ | $50.10_{\uparrow00.35}$ | $48.76_{\downarrow00.63}$ | $\underline{53.54}_{\uparrow03.61}$ | $\underline{41.90}_{\uparrow08.09}$ | $31.76_{\downarrow03.18}$ | $20.99_{\downarrow00.70}$ |
| | FedNova [NeurIPS20] | $36.11_{\uparrow02.84}$ | $34.30_{\uparrow02.33}$ | $32.44_{\uparrow01.77}$ | $32.21_{\uparrow01.98}$ | $53.16_{\uparrow03.41}$ | $51.93_{\uparrow02.54}$ | $46.76_{\downarrow03.17}$ | $31.77_{\downarrow02.04}$ | $35.57_{\uparrow00.63}$ | $21.20_{\downarrow00.49}$ |
| | MOON [CVPR21] | $32.63_{\downarrow00.64}$ | $31.47_{\downarrow00.50}$ | $29.48_{\downarrow01.19}$ | $29.32_{\downarrow00.91}$ | $51.90_{\uparrow02.15}$ | $50.60_{\uparrow01.21}$ | $50.80_{\uparrow00.87}$ | $35.27_{\uparrow01.46}$ | $35.37_{\uparrow00.43}$ | $\underline{22.17}_{\uparrow00.48}$ |
| FGL | FGSSL [IJCAI23] | $36.29_{\uparrow03.02}$ | $34.30_{\uparrow02.33}$ | $32.59_{\uparrow01.92}$ | $32.30_{\uparrow02.07}$ | $56.65_{\uparrow06.90}$ | $55.75_{\uparrow06.36}$ | $38.35_{\downarrow11.58}$ | $19.18_{\downarrow14.63}$ | $37.57_{\uparrow02.63}$ | $19.62_{\downarrow02.07}$ |
| | FedGTA [VLDB24] | $32.54_{\downarrow00.73}$ | $30.82_{\downarrow01.15}$ | $28.59_{\downarrow02.08}$ | $28.82_{\downarrow01.41}$ | $50.00_{\uparrow00.25}$ | $50.82_{\uparrow01.43}$ | $49.15_{\downarrow00.78}$ | $31.57_{\downarrow02.24}$ | $34.73_{\downarrow00.21}$ | $21.24_{\downarrow00.45}$ |
| | FedTAD [IJCAI24] | $31.44_{\downarrow01.83}$ | $30.36_{\downarrow01.61}$ | $31.70_{\uparrow01.03}$ | $29.41_{\downarrow00.82}$ | $51.80_{\uparrow02.05}$ | $51.43_{\uparrow02.04}$ | $44.75_{\downarrow05.18}$ | $33.44_{\downarrow00.37}$ | $35.73_{\uparrow00.79}$ | $21.08_{\downarrow00.61}$ |
| Robust FL | FedNoRo [IJCAI23] | $33.18_{\downarrow00.09}$ | $32.01_{\uparrow00.04}$ | $30.81_{\uparrow00.14}$ | $30.88_{\uparrow00.65}$ | $49.09_{\downarrow00.66}$ | $49.54_{\uparrow00.15}$ | $49.93_{\uparrow00.00}$ | $33.81_{\uparrow00.00}$ | $35.57_{\uparrow00.63}$ | $\mathbf{22.41}_{\uparrow00.72}$ |
| | FedNed [AAAI24] | $35.74_{\uparrow02.47}$ | $33.20_{\uparrow01.23}$ | $32.00_{\uparrow01.33}$ | $31.87_{\uparrow01.64}$ | $46.38_{\downarrow03.37}$ | $44.94_{\downarrow04.45}$ | $53.38_{\uparrow03.45}$ | $37.86_{\uparrow04.05}$ | $32.82_{\downarrow02.12}$ | $21.37_{\downarrow00.32}$ |
| | FedCorr [CVPR22] | $33.91_{\uparrow00.64}$ | $28.59_{\downarrow03.38}$ | $28.00_{\downarrow02.67}$ | $29.52_{\downarrow00.71}$ | $51.52_{\uparrow01.77}$ | $20.39_{\downarrow29.00}$ | $50.73_{\uparrow00.80}$ | $13.46_{\downarrow20.35}$ | $\underline{37.96}_{\uparrow03.02}$ | $17.27_{\downarrow04.42}$ |
| Robust GL | CRGNN [NN24] | $\underline{37.29}_{\uparrow04.02}$ | $\underline{35.87}_{\uparrow03.90}$ | $\underline{37.48}_{\uparrow06.81}$ | $\underline{34.87}_{\uparrow04.64}$ | $\underline{63.59}_{\uparrow13.84}$ | $\underline{58.58}_{\uparrow09.19}$ | $47.08_{\downarrow02.85}$ | $41.74_{\uparrow07.93}$ | $35.35_{\uparrow00.41}$ | $21.40_{\downarrow00.29}$ |
| | RTGNN [WWW23] | $32.91_{\downarrow00.36}$ | $28.21_{\downarrow03.76}$ | $36.44_{\uparrow05.77}$ | $32.83_{\uparrow02.60}$ | $46.89_{\downarrow02.86}$ | $47.82_{\downarrow01.57}$ | $29.36_{\downarrow20.57}$ | $14.64_{\downarrow19.17}$ | $35.88_{\uparrow00.94}$ | $21.34_{\downarrow00.35}$ |
| | CLNode [WSDM23] | $35.47_{\uparrow02.20}$ | $33.41_{\uparrow01.44}$ | $31.41_{\uparrow00.74}$ | $32.10_{\uparrow01.87}$ | $49.60_{\downarrow00.15}$ | $48.98_{\downarrow00.41}$ | $52.72_{\uparrow02.79}$ | $37.96_{\uparrow04.15}$ | $34.12_{\downarrow00.82}$ | $21.96_{\uparrow00.27}$ |
| Robust FGL | HYPERION | $\mathbf{41.50}_{\uparrow08.23}$ | $\mathbf{36.16}_{\uparrow04.19}$ | $\mathbf{43.11}_{\uparrow12.44}$ | $\mathbf{41.11}_{\uparrow10.88}$ | $\mathbf{74.85}_{\uparrow25.10}$ | $\mathbf{74.04}_{\uparrow24.65}$ | $\mathbf{70.10}_{\uparrow20.17}$ | $\mathbf{49.85}_{\uparrow16.04}$ | $\mathbf{39.65}_{\uparrow04.71}$ | $21.72_{\uparrow00.03}$ |

where $\Lambda_c^{-1}$ represents the inverse of the covariance matrix of all prototypes in class $c$, and $\varepsilon$ is used to avoid division by zero errors. The intra-class prototype distance is calculated as:

$$d_{\text{intra}}(z_{m_i}) = \frac{1}{|P_{y_i}| - 1} \sum_{k=1}^{|P_{y_i}|-1} \left[ \frac{1}{(z_{m_i} - p_m^{y_i,k})^\top \Lambda_{y_i}^{-1} (z_{m_i} - p_m^{y_i,k}) + \varepsilon} \right]. \tag{17}$$

Finally, we calculate the comprehensive outlier score and rank all nodes based on their outlier risks, pruning the top $\eta\%$ of the high-risk samples in each training round:

$$\text{Score}(z_{m_i}) = \frac{d_{\text{inter}}(z_{m_i})}{d_{\text{inter}}(z_{m_i}) + d_{\text{intra}}(z_{m_i})} \longrightarrow \mathcal{V}_m^{t+1} = \left\{ z_{m_i} \in \mathcal{V}_m^t \mid \text{rank}(z_{m_i}) > \lfloor \eta * |\mathcal{V}_m^t| \rfloor \right\}, \tag{18}$$

where $\mathcal{V}_m^t$ represents the sample nodes in the $t$ round, and $\eta$ denotes the pruning ratio. Further discussion and limitations can be found in Appendix E and Appendix F.

# 4 Experiment

In this section, we comprehensively evaluate HYPERION through four key axes: **Q1** (Superiority), **Q2** (Resilience), **Q3** (Effectiveness), and **Q4** (Sensitivity).

## 4.1 Experimental Setup

**Datasets.** To effectively evaluate the performance of our approach, we utilize five benchmark graph datasets of various scales and distributions with different characteristics, including Cora [36], CiteSeer [11], PubMed [3], Physics[43], and Amazon_ratings. These datasets represent a wide range of domains and are commonly used in graph-based machine learning tasks. Detailed descriptions and dataset splits for these datasets can be found in Appendix C.1. Furthermore, the implementation details and parameter settings can be found in Appendix C.3.

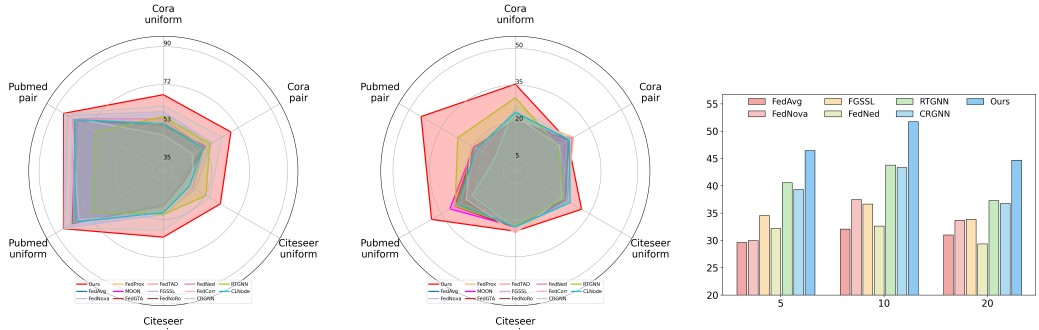

Figure 3: We report the performance of different methods at various noise ratios, with datasets including Cora, Citeseer, and Pubmed, and noise types of uniform and pair. The red color represents the performance of `HYPERION`. *(First)*: 20% mild noise. *(Second)*: 70% severe noise. *(Third)*: We compare `HYPERION` with several FL and FGL methods on the Cora dataset under uniform noise, as the number of clients ranges from 5 to 20.

**Counterparts.** We compare `HYPERION` against several traditional FL methods: (1) **FedAvg** [AS-TAT17] [37], (2) **FedProx** [MLSys20] [25], (3) **FedNova** [NeurIPS20] [54], (4) **MOON** [CVPR 21] [24]; four popular FGL approaches: (5) **FGSSL** [IJCAI23] [16]; (6) **FedTAD** [IJCAI24] [70], (7) **FedGTA** [VLDB24] [28]; three Robust FL methods: (8) **FedNoRo** [IJCAI23] [58], (9) **FedNed** [AAAI24] [33], (10) **FedCorr** [CVPR22] [60];three Robust GraghLearning methods: (11) **CRGNN** [NN24] [27], (12) **RTGNN** [WWW23] [40], (13) **CLNode** [WSDM23] [56]. Detailed descriptions of all the baselines can be found in Appendix C.2.

## 4.2 Superiority

To answer **Q1**, we conducted systematic evaluations in a variety of noise environments, including two typical noise types (uniform noise and pair noise) and three noise intensity levels (0.2 for low noise ratio, 0.5 for medium noise ratio, and 0.7 for high noise ratio). The perfmance results are presented in Tab. 1, and Figure 3. Several observations from these experiments are summarized (**Obs.**):

**Obs. ❶ Existing approaches exhibit suboptimal performance in FGL scenarios with noisy labels.** For instance, in the uniform noise mode with a 50% noise ratio, most previous methods achieve accuracy rates below 40% on the Cora dataset. Notably, under a 70% noise ratio, the performance of most of these methods deteriorates significantly, worsening with average accuracy rates consistently dropping below 25%. Moreover, the existing robust methods do not demonstrate substantial improvements in noisy FGL scenarios, with the performance of most approaches being comparable to that of traditional FL and FGL methods.

**Obs. ❷ `HYPERION` demonstrates remarkable robustness across various noise scales.** Under moderate noise conditions (0.5), `HYPERION` shows a clear and significant advantage. As shown in Tab. 1 (upper), `HYPERION` consistently outperforms both FL and FGL baselines across various datasets and noise types. In the Citeseer-uniform setting, it achieves an accuracy of 47.11%, surpassing the best baseline, RTGNN (41.93%), by 5.18 percentage points. Additionally, as shown in Figure 3 (Second), `HYPERION` consistently outperforms both FL and FGL baselines across various noise scales. It is evident that, under a noise scale of 0.2, `HYPERION` achieves varying degrees of performance improvement over all baselines. In high-noise environments, `HYPERION` also demonstrates an average performance gain of 8.1% to 10.1% compared to the baselines, including RTGNN.

## 4.3 Resilience

To address **Q2**, we evaluate the performance of each method on the Cora dataset under the 0.5 uniform noise setting, across different client scales. Figure 3 (Third) illustrates that `HYPERION` consistently achieves robust performance gains across varying client numbers (5-20), outperforming FedAvg by at least 13.69% while maintaining a minimum 6% advantage over the top-performing baseline method. This demonstrates that `HYPERION` effectively identifies noise and maintains stable performance, even under challenging conditions with varying client populations.

## 4.4 Effectiveness

To address **Q3**, we conduct an ablation study on the key components of our method, both at the client-side and server-side, under a noise scale of 0.5. Tab. 2 reports the performance of HYPERION and its variants by removing specific components from the TP-HSL and HS-CNC modules-namely, the structural loss, robust node selection, and semantic loss. Tab. 3 presents the results for HYPERION and its variants derived from the GA-HSP module, where we ablate server-side distillation (SD), node pruning (NP), and client classification with server distillation (CC+SD). Individually, both TP-HSL and HS-CNC contribute significantly to improving model accuracy. Moreover, GA-HSP demonstrates substantial effectiveness in integrating global reliable information, thereby reinforcing the robustness of our design in mitigating label noise in FGL settings.

Table 2: **Ablation study** of **TP-HSL** and **HS-CNC** on the client-side of HYPERION. All results are reported under 0.5 noise ratio and 10-client scale.

| Client | Cora | | Citeseer | |
|---|---|---|---|---|
| | uniform | pair | uniform | pair |
| w/o $L_{STR}$ | 50.27 | 36.93 | 42.37 | 42.07 |
| w/o $L_{CNC}$ | 50.82 | 35.28 | 43.56 | 38.37 |
| w/o $L_{SEM}$ | 50.09 | 36.20 | 40.29 | 41.04 |
| HYPERION | **56.31** | **41.50** | **47.11** | **43.11** |

Table 3: **Ablation study** of **GA-HSP** on the server-side of HYPERION. w/o CC+SD means SD depends on CC, so without CC, SD is also removed.

| Server | Cora | | Citeseer | |
|---|---|---|---|---|
| | uniform | pair | uniform | pair |
| w/o SD | 49.18 | 37.11 | 39.85 | 38.67 |
| w/o NP | 37.29 | 34.83 | 30.07 | 31.26 |
| w/o CC+SD | 44.66 | 36.40 | 36.07 | 35.70 |
| HYPERION | **56.31** | **41.50** | **47.11** | **43.11** |

## 4.5 Sensitivity

To address **Q4**, we perform sensitivity analyses on hyperparameters of HYPERION. Specifically, we examine the model's performance under varying values of $\lambda$, $\eta$, $\alpha$, and $\beta$, as illustrated in Figure 4, where these hyperparameters are fixed at different scales and values. We systematically vary the hyperparameters $\lambda$ and $\eta$ within the ranges $[0.01, 0.05]$ and $[0.90, 0.98]$, respectively, to evaluate the stability of GA-HSP under different settings. For TP-HSL, we vary $\alpha$ and $\beta$ with in the ranges $[0.4, 0.6]$ and $[0.6, 0.8]$, using a step size of 0.05. The results indicate that the choice of $\lambda$ and $\eta$ has a minimal impact on the performance of HYPERION. However, when $\alpha$ is within the range of $[0.55, 0.6]$ and $\beta$ is within the range of $[0.7, 0.75]$, HYPERION achieves the best performance across all datasets.

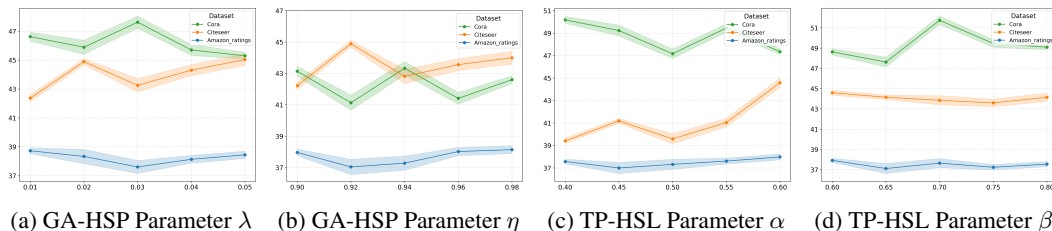

(a) GA-HSP Parameter $\lambda$    (b) GA-HSP Parameter $\eta$    (c) TP-HSL Parameter $\alpha$    (d) TP-HSL Parameter $\beta$

Figure 4: **Analysis on hyper-parameter in** HYPERION. Node classification results under varying values of $\lambda$, $\eta$, $\alpha$, and $\beta$. The colors green, blue, and yellow correspond to performance on the Cora, Amazon_ratings, and Citeseer datasets, respectively. All experiments are conducted using 50%-uniform noise.

We investigate the impact of the hyperparameter $K$ on the performance and efficiency of HYPERION. Specifically, we vary $K \in \{1, 2, 3, 4\}$ in the Citeseer-pair setting and observe the corresponding changes in performance. As shown in Figure 5, setting $K = 1$ results in under-learning of hypersphere, leading to consistently lower performance. In contrast, increasing $K$ to 3 or 4 yields marginal performance improvements. In all experiments, we set $K = 3$.

Figure 5: Performance across varying $K$ values under different noise ratios.

| K | 20% | | 50% | | 70% | |
|---|---|---|---|---|---|---|
| | ACC | F1-M | ACC | F1-M | ACC | F1-M |
| 1 | 59.41 | 52.13 | 38.81 | 36.78 | 20.48 | 20.53 |
| 2 | 59.56 | 52.24 | 39.70 | 37.92 | 21.33 | 21.39 |
| 3 | **61.93** | **56.58** | **43.11** | **41.11** | 24.59 | 24.48 |
| 4 | 61.04 | 55.96 | 38.07 | 36.02 | **24.65** | **25.02** |

## 5    Conclusion

In this work, we propose an innovative exploration of robust FGL in noisy environment. To achieve this goal, we project nodes onto hyperspherical embedding space, thereby introducing a novel framework, `HYPERION`. For robust representation, TP-HSL is employed to project nodes onto hyperspherical space, effectively addressing the coupling problem associated with complex topologies and malicious noise. Leveraging hyperspherical representations, we also introduce HS-CNC, which filters out potential noisy nodes by considering both structural and semantic factors. Specifically, for effective global collaboration, we further design GA-HSP to facilitate knowledge purification. By integrating these three strategies, `HYPERION` outperforms various state-of-the-art methods in node classification tasks across different noisy scenarios.

## Acknowledgement

This work is supported by National Natural Science Foundation of China under Grant (62225113, 623B2080), and the Innovative Research Group Project of Hubei Province under Grants 2024AFA017. The supercomputing system at the Supercomputing Center of Wuhan University supported the numerical calculations in this paper.

# A Notations

We present a comprehensive review of the commonly used notations and their definitions in Tab. 4.

| Notation | Definition |
|---|---|
| $\mathcal{G}_m$ | Graph data for the $m$-th client . |
| $N_m$ | The number of nodes for the $m$-th client. |
| $\mathcal{V}_m$ | The node set of $\mathcal{G}_m$. |
| $\mathcal{X}_m$ | The feature matrix of $\mathcal{G}_m$. |
| $\mathcal{A}_m$ | The adjacency matrix of $\mathcal{G}_m$. |
| $\mathcal{Y}_m$ | The one-hot label matrix of $\mathcal{G}_m$. |
| $\mathcal{X}_{m_i}$ | The feature of node $i$ in $\mathcal{G}_m$. |
| $\mathcal{A}_{m_i}$ | The edges of node $i$ in $\mathcal{G}_m$. |
| $M$ | The number of clients. |
| $d$ | The dimension of the node feature. |
| $C$ | The number of node classes. |
| $K$ | The number of prototypes in each class per client. |
| $\mathcal{V}_m^L$ | The labeled node set of $\mathcal{G}_m$. |
| $\mathcal{V}_m^U$ | The unlabeled node set of $\mathcal{G}_m$. |
| $\mathcal{N}_m^L$ | The number of labeled nodes for the $m$-th client. |
| $\mathcal{N}_m^U$ | The number of unlabeled nodes for the $m$-th client. |
| $f_{\theta_m}$ | The classifier of the $m$-th client. |
| $\hat{\mathcal{Y}}_m$ | The prediction matrix of $\mathcal{G}_m$. |
| $\hat{\mathcal{Y}}_m^N$ | The noisy label matrix of $\mathcal{G}_m$. |
| $\mathcal{L}$ | The loss function. |
| $\epsilon$ | The fixed probability that the true label is flipped to a specific paired class. |
| $z_{m_i}$ | The hyperspherical embedding representation of node $i$ in $\mathcal{G}_m$. |
| $\mathbf{p}_m^{c,k}$ | The $k$-th prototype of the $c$-th class of the $m$-th client. |
| $\omega_m^{c,k}$ | The $k$-th prototype weight of the $c$-th class of the $m$-th client. |
| $\kappa$ | The concentration parameter. |
| $\mathbf{S}_m$ | The similarity metrix between prototypes in the $m$-th client. |
| $\mathbf{S}_m^{pos}$ | The similarity matrix between intra-class prototypes in the $m$-th client |
| $\mathbf{S}_m^{neg}$ | The similarity matrix between inter-class prototypes in the $m$-th client |
| $\Lambda_c$ | The covariance matrix of all prototypes in class $c$. |
| $\eta$ | The pruning ratio of detected nodes. |
| $\phi^{pos}$ | The clean clients set. |
| $\phi^{neg}$ | The malicious clients set. |
| $\alpha$ | The struct loss weight. |
| $\beta$ | The semantic loss weight. |
| $\lambda$ | The hyperparameter controlling the entropy regularization term. |

Table 4: Notation and Definitions.

# B Related work.

**Federated Graph Learning(FGL).** FGL enhances Federated Learning (FL) by extending it to graph-structured data, facilitating decentralized training while safeguarding raw graph data, thereby bolstering privacy protection[14, 19, 30, 4, 53, 52]. Current research primarily focuses on addressing the non-IID data problem in FGL. For instance, FedGCN [61] employs an attention mechanism to dynamically reweight local model parameters, mitigating the impact of data distribution heterogeneity. FGSSL [16] further decomposes the Non-IID issue into node-level semantic divergence and graph-level structural discrepancy, calibrating them separately. However, these approaches overlook a critical practical challenge: label noise caused by annotator negligence or bias may extensively exist in local data, significantly degrading the global model's generalization performance. To address this limitation, we propose a robust FGL framework grounded in hypersphere representation learning, which enhances the model's capacity to capture subtle structural differences in graph data, thereby maintaining stable classification performance under label noise.

**Robust Federated Learning and Graph Learning.** The existing solutions to the noisy label problem in FL can be broadly classified into two categories: label correction and self-supervised learning. The first category involves label correction mechanisms, which reassign noisy labels based on representations extracted from the training data. These include methods like nearest neighbors in the embedding space [48] and predictions from the global model [60]. The second category leverages self-supervised learning to obtain more robust representations, as seen in methods such as RoFL [10] and FedNed [33]. For instance, FedNed [33] reduces the risk of propagating incorrect information by using noisy client negative distillation, while FedDPCont [8] promotes robust learning by randomly selecting contrastive labels and sharing them with the server. Analogous challenges are also prevalent in the domain of graphs, where noisy labels and structural complexities similarly hinder model performance. In recent years, a growing body of research has focused on developing GNN methods tailored for robust graph learning under label noise. Some methods have achieved significant success by incorporating techniques such as loss modulation [39, 66, 29, 9], robust training strategies [56], graph structure augmentation [6, 40, 67], and contrastive learning [64, 27]. However, all these methods are not conducive to a more fine-grained structural learning, leading to the mixing of valid signals and noise components in the feature space, and our method is the first attempt at leveraging hypersphere learning for robust federated graph learning.

## C  Experimental Details.

### C.1  Dataset Details

To assess the effectiveness of `HYPERION`, we conduct experiments on eight real-world graph datasets: Cora, CiteSeer, PubMed, Physics, and Amazon-ratings. Each dataset is split into training, validation, and test sets in a fixed 20%/40%/40% ratio. The key statistics of these datasets are summarized in Tab. 5. A detailed description is provided below:

- **Cora, CiteSeer, and PubMed.** These three citation network datasets are standard benchmarks in graph-based machine learning, especially for tasks like node classification and link prediction. In these datasets, nodes correspond to academic papers, while edges represent citation links. Each node is assigned a class label, and its feature vector is constructed from textual information such as words in the title or abstract. These datasets exhibit sparsity and high dimensionality, making them well-suited for evaluating the effectiveness and scalability of graph neural networks (GNNs).
- **Amazon-ratings.** This dataset is derived from the Amazon product co-purchasing network metadata in the SNAP repository. Nodes represent products (books, music CDs, DVDs, VHS tapes), and edges connect products that are frequently purchased together. The task is to predict the average rating given to a product by reviewers. The authors categorize the possible rating values into five classes. For node features, they use the average of fastText embeddings of the words in the product descriptions. To reduce the size of the graph, only the largest connected component of the 5-core subgraph is considered.
- **Coauthor-Physics.** Coauthor-Physics is an academic network containing co-authorship relationships based on the Microsoft Academic Graph. Nodes in the graph represent authors and edges represent co-authorship relationships. In the dataset, authors are categorized into five classes based on their research areas, and the nodes are characterized as bag-of-words representations of keywords of papers.

| Dataset | #Nodes | #Edges | #Classes | #Features |
|---|---|---|---|---|
| Cora | 2,708 | 5,278 | 7 | 1,433 |
| Citeseer | 3,327 | 4,552 | 6 | 3,703 |
| Pubmed | 19,717 | 44,324 | 3 | 500 |
| Amazon-ratings | 24,492 | 97,050 | 5 | 300 |
| Coauthor-Physics | 34,493 | 530,417 | 5 | 8,415 |

Table 5: **Statistics** of datasets used in experiments.

### C.2  Counterpart Details

This section provides a comprehensive overview of the baseline approaches employed in our study.

- **FedAvg** [ASTAT17]. A foundational algorithm in Federated Learning, FedAvg operates by allowing clients to independently train models on their local datasets and subsequently transmit their model updates to a central server. The server performs a weighted aggregation of these updates to refine the global model, which is then redistributed to the clients for further local training. By transmitting only model parameters instead of raw data, FedAvg reduces communication costs and enhances privacy. However, it struggles with performance degradation in scenarios where client data distributions are highly non-IID [26, 38].
- **FedProx** [MLSys20]. As an enhancement of FedAvg, FedProx is specifically designed to address the challenges posed by statistical heterogeneity in federated learning. It introduces an additional regularization term that constrains local updates, preventing excessive divergence from the global model. This proximal term mitigates the impact of local data distribution shifts, leading to more stable convergence. By ensuring consistency in updates across clients, FedProx demonstrates improved robustness in non-IID settings.
- **FedNova** [NeurIPS20]. FedNova refines the FedAvg framework by introducing normalization to local updates before aggregation. Unlike standard averaging methods, FedNova ensures that each client's contribution to the global model is proportional to the amount of data it possesses. This approach addresses the issue of unequal client influence, leading to more balanced and efficient convergence. FedNova is particularly beneficial in federated environments where data distributions are skewed across clients.
- **FGSSL** [IJCAI23]. FGSSL addresses local client distortion caused by both node-level semantics and graph-level structures. It improves discrimination by contrasting nodes from different classes, aligning local nodes with their global counterparts of the same class while pushing them away from different classes. To handle structural information, it transforms adjacency relationships into similarity distributions and distills relational knowledge from the global model into local models. This approach preserves both structural integrity and discriminability, achieving superior performance on multiple graph datasets.
- **FedTAD** [IJCAI24]. FedTAD addresses subgraph heterogeneity in FL by decomposing local graph variations into label and structural differences, preventing inconsistent model aggregation. It enhances knowledge transfer via topology-aware distillation, boosting FL reliability and efficiency.
- **FedGTA** [VLDB24]. FedGTA is tailored for large-scale graph federated learning, tackling issues of slow convergence and suboptimal scalability. Unlike prior methods that focus on either optimization strategies or complex local models, FedGTA integrates topology-aware local smoothing with mixed neighbor feature aggregation to improve learning efficiency [69]. By leveraging graph structures in aggregation, it enhances scalability and performance in federated graph learning.
- **MOON** [CVPR21]. MOON adopts a model-contrastive approach to address data heterogeneity in federated learning. The framework utilizes similarities between model representations to correct local training through model-level contrastive learning, providing an effective solution for collaborative training with deep learning models on image datasets while preserving data privacy.
- **FedNoRo** [IJCAI23]. FedNoRo adopts a two-stage framework to address class-imbalanced global data with heterogeneous label noise in federated learning. The method first identifies noisy clients through per-class loss indicators and Gaussian Mixture Modeling, then performs noise-robust federated updates via joint knowledge distillation and distance-aware aggregation, specifically designed for realistic medical scenarios with data imbalance and complex noise patterns.
- **FedNed** [AAAI24]. FedNed adopts a negative distillation framework to effectively leverage extremely noisy clients in federated learning. The method first identifies noisy clients, then innovatively utilizes them as 'bad teachers' through a dual-training approach: one model trained on original noisy labels for reverse knowledge distillation, and another on global model-generated pseudo-labels for conditional participation in aggregation. This approach transforms noisy clients from detrimental elements into valuable contributors while progressively enhancing their trustworthiness through pseudo-label refinement
- **FedCorr** [CVPR22]. FedCorr adopts a multi-stage framework to address heterogeneous label noise in federated learning while preserving data privacy. The method first dynamically identifies noisy clients through model prediction subspace analysis and per-sample loss evaluation, then employs an adaptive local proximal regularization to handle data heterogeneity. After fine-tuning on clean clients and correcting labels for noisy ones, FedCorr performs final training across all clients to fully utilize available data, effectively handling varying noise levels without requiring prior assumptions about client noise models.
- **CRGNN** [NN24]. CRGNN addresses label noise in GNNs by combining neighborhood-based label correction and contrastive learning. It utilizes message passing neural networks to update

| Category | Methods | 20% Label Noise | | | | | | 70% Label Noise | | | | | |
|---|---|---|---|---|---|---|---|---|---|---|---|---|---|
| | | Cora | | CiteSeer | | PubMed | | Cora | | CiteSeer | | PubMed | |
| | | Uniform | Pair | Uniform | Pair | Uniform | Pair | Uniform | Pair | Uniform | Pair | Uniform | Pair |
| FL | FedAvg [ASTAT17] | $48.00_{\uparrow00.00}$ | $48.41_{\uparrow00.00}$ | $48.57_{\uparrow00.00}$ | $45.00_{\uparrow00.00}$ | $80.18_{\uparrow00.00}$ | $77.08_{\uparrow00.00}$ | $18.34_{\uparrow00.00}$ | $21.31_{\uparrow00.00}$ | $23.24_{\uparrow00.00}$ | $23.53_{\uparrow00.00}$ | $24.56_{\uparrow00.00}$ | $19.29_{\uparrow00.00}$ |
| | FedNova [NeurIPS20] | $53.50_{\uparrow05.50}$ | $53.47_{\uparrow05.06}$ | $44.15_{\downarrow04.42}$ | $49.05_{\uparrow04.05}$ | $81.71_{\uparrow01.53}$ | $79.86_{\uparrow02.78}$ | $21.89_{\uparrow03.55}$ | $18.12_{\downarrow03.19}$ | $23.45_{\uparrow00.21}$ | $22.03_{\downarrow01.50}$ | $24.24_{\downarrow00.32}$ | $17.43_{\downarrow01.86}$ |
| | FedProx [MLSys20] | $52.25_{\uparrow04.25}$ | $53.76_{\uparrow05.35}$ | $39.64_{\downarrow08.93}$ | $46.20_{\uparrow01.20}$ | $79.61_{\downarrow00.57}$ | $78.00_{\uparrow00.92}$ | $21.05_{\uparrow02.71}$ | $\mathbf{23.26}_{\uparrow01.95}$ | $24.04_{\uparrow00.80}$ | $\mathbf{25.28}_{\uparrow01.75}$ | $24.11_{\downarrow00.45}$ | $18.81_{\downarrow00.48}$ |
| | MOON [CVPR21] | $48.51_{\uparrow00.51}$ | $49.29_{\uparrow00.88}$ | $38.75_{\downarrow09.82}$ | $45.12_{\uparrow00.12}$ | $79.33_{\downarrow00.85}$ | $78.34_{\uparrow01.26}$ | $18.28_{\downarrow00.06}$ | $20.62_{\downarrow00.69}$ | $22.43_{\downarrow00.81}$ | $21.81_{\downarrow01.72}$ | $21.64_{\downarrow02.92}$ | $19.05_{\downarrow00.24}$ |
| FGL | FedGTA [VLDB24] | $49.00_{\uparrow01.00}$ | $48.60_{\uparrow00.19}$ | $37.32_{\downarrow11.25}$ | $45.16_{\uparrow00.16}$ | $79.69_{\downarrow00.49}$ | $76.59_{\downarrow00.49}$ | $18.93_{\uparrow00.59}$ | $21.75_{\uparrow00.44}$ | $22.18_{\downarrow01.06}$ | $23.20_{\downarrow00.33}$ | $\underline{26.43}_{\uparrow01.87}$ | $19.46_{\uparrow00.17}$ |
| | FedTAD [IJCAI24] | $47.72_{\downarrow00.28}$ | $47.61_{\downarrow00.80}$ | $38.02_{\downarrow10.55}$ | $44.36_{\downarrow00.64}$ | $79.97_{\downarrow00.21}$ | $78.04_{\uparrow00.96}$ | $20.29_{\uparrow01.95}$ | $21.14_{\downarrow00.17}$ | $21.10_{\downarrow02.14}$ | $22.30_{\downarrow01.23}$ | $19.07_{\downarrow05.49}$ | $19.40_{\uparrow00.11}$ |
| | FGSSL [IJCAI23] | $54.70_{\uparrow06.70}$ | $53.33_{\uparrow04.92}$ | $45.71_{\downarrow2.86}$ | $48.54_{\uparrow03.54}$ | $84.81_{\uparrow04.63}$ | $\underline{83.35}_{\uparrow06.27}$ | $21.75_{\uparrow03.41}$ | $20.04_{\downarrow01.27}$ | $22.25_{\downarrow00.00}$ | $20.42_{\downarrow03.11}$ | $11.51_{\downarrow13.05}$ | $13.60_{\downarrow05.69}$ |
| Robust FL | FedNoRo [IJCAI23] | $47.86_{\downarrow00.14}$ | $48.37_{\downarrow00.04}$ | $38.79_{\downarrow09.78}$ | $44.88_{\downarrow00.12}$ | $80.12_{\downarrow00.06}$ | $77.44_{\uparrow00.36}$ | $18.34_{\uparrow00.00}$ | $21.30_{\downarrow00.01}$ | $22.77_{\downarrow00.47}$ | $23.35_{\downarrow00.18}$ | $25.01_{\uparrow00.45}$ | $18.87_{\downarrow00.42}$ |
| | FedNed [AAAI24] | $51.63_{\uparrow03.63}$ | $48.30_{\downarrow00.11}$ | $43.19_{\downarrow05.38}$ | $48.99_{\uparrow03.99}$ | $78.86_{\downarrow01.32}$ | $79.60_{\uparrow02.52}$ | $19.05_{\uparrow00.71}$ | $21.26_{\downarrow00.05}$ | $24.48_{\uparrow01.24}$ | $23.71_{\downarrow00.18}$ | $25.06_{\uparrow00.50}$ | $18.44_{\downarrow00.85}$ |
| | FedCorr [CVPR22] | $35.94_{\downarrow12.06}$ | $38.78_{\downarrow09.63}$ | $47.55_{\downarrow01.02}$ | $37.32_{\downarrow07.68}$ | $70.45_{\downarrow09.73}$ | $71.80_{\downarrow05.28}$ | $19.70_{\uparrow01.36}$ | $18.70_{\downarrow02.61}$ | $20.21_{\downarrow03.03}$ | $24.51_{\uparrow00.98}$ | $11.27_{\downarrow13.29}$ | $21.59_{\uparrow02.30}$ |
| Robust GL | CRGNN [NN24] | $58.15_{\uparrow10.15}$ | $\underline{61.98}_{\uparrow13.57}$ | $48.30_{\downarrow00.27}$ | $53.42_{\uparrow08.42}$ | $\underline{84.41}_{\uparrow04.23}$ | $82.81_{\uparrow05.73}$ | $\underline{24.81}_{\uparrow06.47}$ | $22.99_{\uparrow01.68}$ | $\underline{26.11}_{\uparrow02.87}$ | $22.95_{\downarrow00.58}$ | $11.27_{\downarrow13.29}$ | $12.07_{\downarrow07.22}$ |
| | RTGNN [WWW23] | $50.14_{\uparrow02.14}$ | $43.06_{\downarrow05.35}$ | $\mathbf{53.48}_{\uparrow04.91}$ | $\underline{53.53}_{\uparrow08.53}$ | $83.35_{\uparrow03.17}$ | $82.59_{\uparrow05.51}$ | $18.99_{\uparrow00.65}$ | $14.68_{\downarrow06.63}$ | $19.30_{\uparrow03.94}$ | $14.50_{\downarrow09.03}$ | $11.82_{\downarrow12.74}$ | $\underline{26.15}_{\uparrow06.86}$ |
| | CLNode [WSDM23] | $49.62_{\uparrow01.62}$ | $49.95_{\uparrow01.54}$ | $43.64_{\downarrow04.93}$ | $48.51_{\uparrow03.51}$ | $78.33_{\downarrow01.85}$ | $78.58_{\uparrow01.50}$ | $20.47_{\uparrow02.13}$ | $19.78_{\downarrow01.53}$ | $22.22_{\downarrow01.02}$ | $22.91_{\downarrow00.62}$ | $25.83_{\uparrow01.27}$ | $17.85_{\downarrow01.44}$ |
| Robust FGL | HYPERION | $\mathbf{62.18}_{\uparrow14.18}$ | $\mathbf{64.67}_{\uparrow16.26}$ | $\underline{53.10}_{\uparrow04.53}$ | $\mathbf{54.47}_{\uparrow09.47}$ | $\mathbf{85.35}_{\uparrow05.17}$ | $\mathbf{85.47}_{\uparrow08.39}$ | $\mathbf{29.73}_{\uparrow11.39}$ | $\underline{23.05}_{\uparrow01.74}$ | $\mathbf{27.19}_{\uparrow03.95}$ | $\underline{24.59}_{\uparrow01.06}$ | $\mathbf{30.67}_{\uparrow06.11}$ | $\mathbf{31.12}_{\uparrow11.83}$ |

Table 6: **Comparison with the state-of-the-art methods on three selected real-world datasets.** The noise is set to **20%** and **70%**, and the number of clients $M$ is set to 10 throughout all experiments. The best and second-best results are highlighted with **bold** and underline, respectively.

node representations, integrating graph contrastive learning for consistent representations across augmented graph views. Finally, CGNN employs an MLP for prediction distributions and iteratively corrects noisy labels by comparing them with their neighbors and choosing the most labels.

- **RTGNN** [WWW23]. RTGNN proposes a noise governance framework that combines self-reinforcement supervision for noisy label correction and consistency regularization to prevent overfitting. The method categorizes labels into clean and noisy types, then applies adaptive supervision by rectifying inaccurate labels and generating pseudo-labels for unlabeled nodes, enabling effective learning from clean labels while mitigating noise impact.
- **CLNode** [WSDM24]. CLNode adopt a curriculum learning strategy to mitigate the impact of label noise. To be specific, it first utilize a multi-perspective difficulty measurer to accurately measure the quality of training nodes. Then employ a training scheduler that selects appropriate training nodes to train GNN in each epoch based on the measured qualities. The authors demonstrated this method enhances the robustness of backbone GNN to label noise.

### C.3 Implementation Details.

The experiments are conducted using NVIDIA GeForce RTX 4090 GPUs as the hardware platform, coupled with Intel(R) Xeon(R) Platinum 8336C CPU @ 2.30GHz. The deep learning framework employed was Pytorch, version 2.5.1, alongside CUDA version 12.2. Our network features a four-layer GCN backbone with uniform 384-dimensional hidden layers throughout the first three layers, each employing symmetric normalization (normalize=True) and ReLU activation, followed by 0.2 dropout for regularization. The final GCN layer produces compact 32-dimensional graph embeddings without activation. These embeddings are processed through a two-layer MLP head with ReLU activation in the hidden layer. The architecture optionally incorporates prototype learning with configurable parameters: each class maintains multiple 32-dimensional prototype vectors, and the prototype contrastive loss operates with a temperature parameter $\tau = 0.07$ to control the similarity distribution sharpness. All GCN layers implement symmetric normalization (normalize=True), and consistent dropout ($p = 0.2$) is applied after each intermediate layer to prevent overfitting. TP-HSL parameter $\alpha$ is set in the range $\{0.40, 0.50, 0.60\}$, $\beta$ Is set in the range $\{0.65, 0.70, 0.75\}$. As for GA-SHP parameter $\lambda$ and $\eta$, we set $\lambda$ in the range $\{0.03, 0.04\}$, $\eta$ in the range $\{0.92, 0.94, 0.96\}$. The number of communication rounds is 100 for all methods. The number of clients $M$ is set to 10 throughout all experiments, except for Figure 3 (Third).

# D  Additional Experimental Results.

We place additional F1-macro score results under 0.2 and 0.7 noisy label ratios in Tab. 6.

# E  Broader Impact.

Our work is an important step in overcoming the widespread and imperceptible labeling noise in FGL. This approach can effectively enhance the topological attention of the model to discriminate the

noise. This could lead to more robust and trustworthy graph learning systems in real-world federated environments, where data quality and consistency are often difficult to guarantee.

# F    Discussion on Limitations.

Although `HYPERION` has demonstrated significant success in efficiently capturing subtle topological differences between nodes of the same class and mitigating malicious noise through a hyperspherical representation, it still faces some limitations. Specifically, our current formulation primarily addresses class label noise, while other noise types (e.g., feature noise or adversarial edge perturbations) may require additional mechanisms beyond the proposed purification framework.

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
