# OpenReview forum: "HYPERION: Fine-Grained Hypersphere Alignment for Robust Federated Graph Learning"
_NeurIPS.cc/2025/Conference — NeurIPS 2025 spotlight_

### Official Review · Reviewer_XQTy · 2025-06-20

**Clarity:** 3
**Significance:** 3
**Originality:** 3
**Rating:** 5
**Confidence:** 4

**Summary:**

This study tackles challenges such as non-IID graph data distribution and adversarial noise perturbation in federated graph learning. It introduces a robust strategy where clients collaboratively learn hyperspherical embeddings to enforce geometric consistency across heterogeneous local graphs, mitigating feature skew. To address the classic differences in client data distribution in federated learning, the approach incorporates Geometric-Aware Hyperspherical Purification. Lastly, adequate experiments are conducted to verify the model effectiveness.

**Questions:**

1. The author did not introduce how to create enhanced views in HS-CNC.
2. Why did the author project the nodes onto the hypersphere? Can the author explain the benefits of doing so?

**Ethical Concerns:**

["NO or VERY MINOR ethics concerns only"]

**Final Justification:**

My initial review was generally correct, and the authors addressed my concerns well, so I am leaving my original score unchanged.

**Limitations:**

My complete set of queries and concerns is captured in the 'Weaknesses' and 'Questions' part.

**Paper Formatting Concerns:**

The submission is well-formatted and meets all style requirements.

**Quality:**

4

**Strengths And Weaknesses:**

Strengths:

1. It is novel and meaningful to propose a robust FGL approach. This study precisely identifies the noise challenges of FGL and tackles them effectively. The overall approach is well-motivated, and the motivation is also explained with clarity.

2. The figures of problem and framework illustration are clear and detailed, and the equations and explanations are reasonable. I believe this is a comprehensive framework for the graph data.

3. The approach proposed in the study is consistent and natural. The key modules and their components are innovative and logically coherent. I believe this approach is inspiring.

Weaknesses:

1. The work lacks analysis of model performance with different local training epochs. Can the author supplement the training effects of his method under different training rounds?

2. It would be better if the reasons for the selection of comparison algorithms and datasets were explained.

---

> ### Author Rebuttal · Authors · 2025-07-30
>
> Thank you for the feedback. Regarding your questions, we would like to clarify the following points:
>
> > `Weaknesses 1`: Add analysis of model performance across different local training epochs/rounds.
>
> We greatly appreciate these insightful suggestions.We have conducted additional experiments to analyze the performance of our method under different local training epochs. Specifically, we varied the number of local training epochs and observed how this impacted the overall model performance:
>
> | Acc      | 25        | 50        | 75        | 100       |
> | -------- | --------- | --------- | --------- | --------- |
> | FedAvg   | 38.48     | 34.1      | 33.27     | 32.91     |
> | MOON     | 38.48     | 35.47     | 33.76     | 33.64     |
> | FedNed   | 34.64     | 34.10     | 33.82     | 32.82     |
> | CRGNN    | 49.27     | 46.98     | 45.78     | 44.61     |
> | HYPERION | **55.24** | **54.34** | **53.75** | **53.56** |
>
> We report Acc values on Cora dataset with 50%-uniform noise. The results clearly show that HYPERION consistently outperforms the other methods across all local training epochs. As the number of epochs increases, we observe a slight decline in performance for all methods, which is typical due to the overfitting risk and the noise in the dataset. However, HYPERION maintains a significant margin over the competing methods even at higher epoch counts, indicating its robustness to noisy labels and its ability to generalize better across different training configurations. We will include more detailed analyses in the revised version.
>
> > `Weaknesses 2`: Justify the selection of comparison algorithms and datasets.
>
> We compare our method, HYPERION, with several state-of-the-art federated learning (FL) and federated graph learning (FGL) methods to comprehensively evaluate its robustness and effectiveness under noisy conditions. These comparison methods include FedAvg, FedProx, FedNova, MOON, and others, which are widely recognized in the FL and FGL communities. We included them because they represent the current standards in federated learning, especially in settings with label noise.Additionally, Robust FL methods like FedNoRo, FedNed, and FedCorr are also included to evaluate how HYPERION compares to methods specifically designed to address noise in federated settings. In addition, we also considered several robust graph learning algorithms, which have demonstrated excellent robustness in the presence of noisy labels in graph data, but have not addressed issues such as the client-side limitations in federated learning.The comparison with these algorithms helps us demonstrate HYPERION’s superior robustness in scenarios where noise is a major challenge.
>
> As for the choice of datasets, we used five benchmark datasets: Cora, CiteSeer, PubMed, Physics, and Amazon_ratings. These datasets were selected because they represent a range of real-world graph data types with varying complexities and noise profiles.
> Cora, CiteSeer, and PubMed are well-established citation networks commonly used in graph learning tasks, and they exhibit challenges such as node sparsity and high feature dimensionality, making them suitable for testing the robustness of our method under label noise.The Amazon_ratings dataset introduces a real-world, user-item co-purchase graph, which has more complex structural properties and is frequently used to benchmark collaborative filtering algorithms. This dataset helps us evaluate how HYPERION performs in scenarios involving collaborative data.The Coauthor-Physics dataset provides a more specialized network of co-author relationships, testing HYPERION’s ability to handle graphs from specific academic domains, demonstrating its applicability to different types of graph-structured data.The details of the datasets are shown in the table below.
>
> | Dataset          | Nodes  | Edges   | Classes | Features |
> | ---------------- | ------ | ------- | ------- | -------- |
> | Cora             | 2,708  | 5,278   | 7       | 1433     |
> | Citeseer         | 3,327  | 4,552   | 6       | 3703     |
> | Pubmed           | 19,717 | 44,324  | 3       | 500      |
> | Amazon-ratings   | 24,492 | 97,050  | 5       | 300      |
> | Coauthor-Physics | 34,493 | 530,417 | 5       | 8,415    |
>
> We sincerely thank you for this suggestion. We will enhance the description of algorithms and datasets in the revised version.
>
> > `Questions 1`: Clarify the enhanced view generation process in HS-CNC.
>
> In the HS-CNC module, enhanced views are created through a dual perturbation strategy that combines structural and feature-level augmentations to identify noisy nodes. Following prior work [1, 2], our method first applies random edge dropping to the adjacency matrix with a predefined probability, disrupting local graph connectivity while preserving global structure. Simultaneously, it randomly masks a portion of node features to simulate semantic variations, generating perturbed versions of the original graph data. These augmented views are then processed through the same model to obtain hyperspherical embeddings, where nodes demonstrating consistent cluster assignments across different perturbed views are retained as reliable samples while inconsistent nodes are filtered as potential noise. This approach effectively amplifies the difference between stable clean nodes and unstable noisy nodes by evaluating their robustness to controlled perturbations, enabling more accurate noise identification without requiring explicit label correction. The view generation process is designed to maintain the intrinsic graph properties while introducing sufficient variation to test node stability, striking a balance between preserving meaningful patterns and exposing noise-sensitive regions of the data.
>
> > `Questions 2`: Explain the advantages of hyperspherical node projection in the method.
>
> By constraining embeddings to a unit sphere, the model leverages angular distance (cosine similarity) rather than Euclidean distance, which naturally amplifies inter-class separation while compacting intra-class clusters. This geometric property makes the model more sensitive to meaningful semantic relationships while being less susceptible to noise and feature magnitude variations. The hyperspherical space enables more stable prototype learning through von Mises-Fisher distributions, facilitates reliable federated aggregation via directional consistency, and provides a principled framework for noise purification through angular outlier detection. Compared to traditional Euclidean spaces, the hyperspherical projection offers superior noise robustness by encoding semantic similarity in angular relationships rather than absolute positions, which is particularly crucial for handling label noise and structural variations in decentralized graph data. This approach allows the model to better capture fine-grained topological differences while maintaining strong generalization across clients.
>
> We sincerely thank you for this suggestion. We will enhance the description of hyperspherical node projection in the revised version.
>
> We hope this clarifies your questions.
>
> [1]: Deep graph contrastive representation learning. In ICML, 2020.
>
> [2]: Contrastive learning of graphs under label noise. In Neural Networks, 2024.

---

> > ### Comment · Reviewer_XQTy · 2025-08-03
> >
> > Thank you for the detailed rebuttal. I have carefully reviewed the authors' responses, and my concerns have been adequately addressed. Accordingly, I will maintain my original score.

---

> > > ### Author Response · Authors · 2025-08-07
> > >
> > > We would like to express our heartfelt thanks for your consistent support. Your positive assessment of our work is greatly appreciated. Please feel free to reach out if you have any additional thoughts or suggestions.
> > >
> > > Thank you for your consideration and valuable time.

---

### Official Review · Reviewer_MNA6 · 2025-06-24

**Clarity:** 3
**Significance:** 3
**Originality:** 4
**Rating:** 5
**Confidence:** 4

**Summary:**

HYPERION proposes a federated graph learning framework that handles noisy labels and complex topologies using hyperspherical embeddings. Its multi-prototype design captures fine-grained structures, while noise calibration and purification (via Wasserstein distillation and Mahalanobis-based methods) enhance robustness. Experiments show strong performance across datasets and noise levels, outperforming existing methods.

**Questions:**

See Strengths And Weaknesses.

**Ethical Concerns:**

["NO or VERY MINOR ethics concerns only"]

**Final Justification:**

I have carefully read the author's response and the other reviewers' comments. The rebuttal has addressed the major concerns raised during the initial review phase. Therefore, I maintain my original score of accept.

**Limitations:**

In the experiment, the authors compared several robust graph learning methods, but they did not explain how to apply these methods in a federated learning scenario.I hope the authors will introduce how they applied these methods to the federal framework.

**Paper Formatting Concerns:**

None.

**Quality:**

4

**Strengths And Weaknesses:**

Strengths:

1. This paper points out several main reasons why noise problems in FGL are difficult to solve. Subtle differences in graph structures and the classic differences in client data distribution in federated learning indeed make it difficult to effectively identify noise. The problem illustration clearly shows this phenomenon. The author's writing and illustrations made it easy for me to understand his intentions.

2. The proposed method of robust FGL appears interesting and innovative. The paper provides a detailed explanation of the methodology. The paper considers the difference between normal topological variations and subtle noise, introducing Topological Prototypes Hyperspherical Learning to fully capture the rich topological differences between nodes of the same class. I find this approach inspiring.

3. This paper conducted comprehensive experiments, compared multiple methods, and provided a persuasive explanation of the experimental results. The proposed method makes sense for the problem. The node classification task is widely used as a downstream task in FGL methods and is practically applicable. Overall, the approach appears to be both methodologically sound and relevant to real-world scenarios.


Weaknesses:

1. On page 9, the author discusses the impact of different numbers of prototypes in prototype clusters on performance under different noise conditions. However, the author does not introduce the experimental results under the F1-macro metric, nor does he provide information on other hyperparameters besides K for this experiment.

2. The TP-HSL component of your approach introduces multiple prototypes per class. Why does this method enable fine-grained structural learning?

---

> ### Author Rebuttal · Authors · 2025-07-30
>
> Thank you for the feedback. We address your concerns below and hope our responses will help update your score:
>
> > `Weaknesses 1`: Add F1-macro results and full hyperparameter details for prototype clusters (p9).
>
> Thank you for your constructive feedback regarding the presentation of experimental results. We appreciate your suggestion to include the F1-macro metric and provide more details about the hyperparameters used in the experiment.We have now included the F1-macro results for the experiment involving different numbers of prototypes in prototype clusters under various noise conditions.The experimental results are shown in the table below.
>
> | F1-macro | 20%       | 50%       | 70%       |
> | -------- | --------- | --------- | --------- |
> | K=1      | 52.13     | 36.78     | 20.53     |
> | K=2      | 52.24     | 37.92     | 21.39     |
> | K=3      | **56.58** | **41.11** | 24.48     |
> | K=4      | 55.96     | 36.02     | **25.02** |
>
> From the perspective of F1-macro, it has been confirmed once again that setting *K* = 1 results in under-learning of hypersphere, leading to consistently lower performance.In contrast, increasing *K* to 3 or 4 yields marginal performance improvements.
>
> We also understand the importance of clarifying the impact of other hyperparameters on the experiment. Specifically, we split the dataset into training, validation, and test sets in a 20%, 40%, and 40% ratio. We used the Louvain partitioner to induce data skew. The experiments were run for 100 rounds, involving 10 clients. Each client was trained for 3 local epochs with a batch size of 24. The learning rate was set to 0.005, and we introduced pairwise label noise to simulate real-world imperfections. Our network features a four-layer GCN backbone with uniform 384-dimensional hidden layers throughout the first three layers,each employing symmetric normalization (normalize=True) and ReLU activation, followed by 0.2 dropout for regularization. The final GCN layer produces compact 32-dimensional graph embeddings without activation. These embeddings are processed through a two-layer MLP head with ReLU activation in the hidden layer. The architecture optionally incorporates prototype learning with configurable parameters: each class maintains multiple 32-dimensional prototype vectors, and the prototype contrastive loss operates with a temperature parameter *τ* = 0*.*07 to control the similarity distribution sharpness. All GCN layers implement symmetric normalization (normalize=True), and consistent dropout (*p* = 0*.*2) is applied after each intermediate layer to prevent overfitting.
>
> We acknowledge the value of testing on diverse systems and will include more comprehensive results including F1-macro in the revised version.
>
> > `Weaknesses 2`: Clarify how TP-HSL's multi-prototype design enables fine-grained structural learning (e.g., through prototype diversity or cluster separation).
>
> In our TP-HSL method, we utilize the von Mises-Fisher (vmF) distribution to model the distribution of node embeddings. The vmF distribution is well-suited for accurately measuring the angular differences between embeddings, especially in high-dimensional spaces. In graph data, node embeddings are typically high-dimensional, and the vmF distribution, through its concentration parameter, effectively quantifies the similarity and dissimilarity between node embeddings. This angular difference measure not only helps capture distributional differences between classes but also provides a deeper understanding of each node's local structural characteristics in the high-dimensional space. In particular, in graphs with complex topological structures, the vmF distribution offers a precise and efficient way to assess subtle differences between nodes.
>
> Additionally, traditional graph learning methods often assign a single prototype to each class, which overlooks the structural variations within the nodes of the same class. In our TP-HSL method, we design multiple prototypes for each class, which help capture the diversity among nodes within the same class in the high-dimensional space. Each prototype represents a distinct topological feature within the class, allowing the class to be partitioned into multiple subspaces, each represented by one prototype. This approach ensures that the node embeddings within each class are better aligned with the corresponding prototypes, thereby improving both intra-class cohesion and inter-class distinction. By doing so, we can finely capture the structural differences within each class.
>
> We sincerely thank you for this suggestion. We will enhance the description of TP-HSL in the revised version.
>
> > `Limitations`: Clarify how the compared robust graph learning methods were adapted for federated learning in the framework.
>
> In our experiments, each robust GL baseline (e.g., CRGNN, RTGNN, CLNode) was applied independently on each client. Specifically:
>
> **Client-Side:** On each client m, the robust GL method was employed to train a local model using the client’s private graph data G_{m}. The original noise-resilience mechanisms proposed in these methods—such as label correction, contrastive learning, or topology-aware regularization—were preserved and applied locally without modification.
>
> **Server-Side:** After local training, model updates from clients were aggregated on the server using the standard FedAvg algorithm. No changes were made to the server-side aggregation logic for these baselines, ensuring consistent treatment across all compared methods.We selected FedAvg as the aggregation strategy due to its simplicity, stability, and widespread use in FL research. More importantly, using a uniform aggregation protocol eliminates the influence of advanced server-side techniques, allowing us to isolate and fairly compare the effectiveness of each robust GL metho’s local training design under federated conditions.
>
> We sincerely thank you for this suggestion. We will enhance the description of implementation details in the revised version.
>
> We hope this clarifies your questions.

---

> > ### Comment · Reviewer_MNA6 · 2025-08-05
> > **Maintaining Original Accept Recommendation**
> >
> > I appreciate the authors’ thorough response and the clarifications provided. Having considered the response and the other reviews, I would like to keep my original score of accept.

---

> > > ### Author Response · Authors · 2025-08-07
> > >
> > > Thank you for your positive feedback and for confirming that our rebuttal has addressed your questions.
> > >
> > > We are truly grateful for your constructive comments throughout the review process. We also want to express our sincere appreciation for your strong support and high rating of our work.
> > >
> > > Thank you again for your time and valuable contribution.

---

### Official Review · Reviewer_kQxL · 2025-06-26

**Clarity:** 3
**Significance:** 4
**Originality:** 4
**Rating:** 4
**Confidence:** 3

**Summary:**

This work introduces HYPERION, a robust federated graph learning framework designed to handle complex topologies and resist malicious noise by leveraging hyperspherical embeddings.
Clients embed local node samples onto a hypersphere with class prototypes and generate perturbed views to identify noise-sensitive regions. During global training, a Gaussian Mixture Model filters clean clients. Prototypes from these trusted clients are aggregated and distilled into a global hypersphere, while biased nodes are pruned to ensure model integrity. Experiments confirm that HYPERION outperforms existing methods and shows remarkable resilience to severe label noise.

**Questions:**

Insufficient methodological justification. The authors employ Mahalanobis distance distillation in GA-HSP but do not provide a thorough rationale for this choice. A more detailed discussion comparing it with alternative distance metrics (e.g., standard L2 distance) would strengthen the methodological foundation.

**Ethical Concerns:**

["NO or VERY MINOR ethics concerns only"]

**Final Justification:**

After rebuttal, some of my concerns have been addressed. overall, this paper is novel, and I will keep the rating.

**Limitations:**

I have put all my doubts in Weaknesses and Questions.

**Paper Formatting Concerns:**

No formatting issues were found. The manuscript adheres to the conference guidelines.

**Quality:**

3

**Strengths And Weaknesses:**

Strengths:
S1: Novel and well-motivated. The authors identify the limitations of coarse-grained representations and the neglect of intra-class topological differences in existing robust FGL methods. It innovatively proposes using hyperspherical embeddings to capture fine-grained structural patterns, providing an approach to mitigate noise propagation.
S2: Effective and thought-provoking. The introduction of multiple topological prototypes per class allows the model to capture subtle structural variations within the same class, a crucial aspect of graph data that is often overlooked. This design significantly heightens the ability to differentiate noisy signals from valid ones. Good job!
S3: Comprehensive framework. The interplay between local noise calibration and global geometric-aware purification creates a holistic defense mechanism against label noise . Furthermore, there has been little work done in this area previously, and the author has indeed solved the problem.
S4: Strong empirical performance. The experimental validation is extensive and compelling. HYPERION consistently and significantly outperforms a wide array of state-of-the-art methods across multiple datasets and under various noise conditions.

Weaknesses:
W1: The authors use Mahalanobis distance when pruning and Wasserstein distance when distilling after clustering, but did not explain in detail the characteristics of these two distances and the differences between them.
W2: The authors compare the proposed approach with several traditional FL/FGL and robust FL and GL methods, but additional comparisons with more FGL and robust FL and GL methods could strengthen this study.

---

> ### Author Rebuttal · Authors · 2025-07-30
>
> Thank you for the feedback. We address your concerns below and hope our responses will help update your score:
>
> > `Weaknesses 1`: Clarify Mahalanobis vs. Wasserstein distance characteristics.
>
> We appreciate the opportunity to elaborate on these distance metrics and their respective roles in our approach.
>
> 1. The **Mahalanobis distance** is used to measure the distance between a point and a distribution, taking into account the covariance structure of the data. By introducing the inverse of the covariance matrix, Mahalanobis distance can automatically adjust the importance of each dimension, avoiding certain dimensions dominating the distance calculation due to scale differences.In geometry, Mahalanobis distance transforms the data space into a standardized space, making distance calculations more equitable.We identify and remove noise nodes that deviate from the normal distribution by calculating the Mahalanobis distance between node embeddings and the global prototype distribution (as shown in Formula 18). This method effectively filters out noise that is not detected by clients due to local perspective limitations.
> 2. The **Wasserstein distance**, measures the minimum cost of transforming one probability distribution into another. It is particularly effective when comparing distributions that are not necessarily aligned or of the same type, making it well-suited for our distillation phase after clustering. By minimizing Wasserstein distance, the probability mass is effectively reallocated between proximate dimensions in the feature space, thereby naturally reinforcing the correlation of benign features between global prototypes, while effectively diminishing the impact of anomalous features.Unlike Mahalanobis distance, which focuses on individual point-to-point distances, Wasserstein distance captures the "flow" of information between distributions, making it ideal for distilling information from the clustered nodes while preserving their relational structure.
>
> We will include more detailed analyses in the revised version.
>
> > `Weaknesses 2`: Add comparisons to more FGL and robust FL/GL methods.
>
> We agree that expanding the set of methods for comparison would provide a more comprehensive evaluation of our proposed approach. We have conducted additional experiments and included comparisons with a broader set of FGL and robust FL/GL methods. These additional comparisons further strengthen our study and provide a more detailed analysis of the performance of our proposed approach in different settings. The newly added experiments include comparisons with methods FedDC, FedDyn, FedProto, Scaffold and Co-teaching, which we believe are highly relevant to the challenges addressed in our work. The noise type is set to 50%-uniform and we report accuracy (%) and F1-macro (%).
>
> New Experimental Results:
>
> | Acc         | Cora      | Citeseer  | Pubmed    | Physics   | Amazon-ratings |
> | ----------- | --------- | --------- | --------- | --------- | -------------- |
> | FedDC       | 45.25     | 37.48     | 52.73     | 50.73     | 36.43          |
> | FedDyn      | 46.53     | 32.15     | 74.34     | 72.52     | 36.28          |
> | FedProto    | 40.22     | 32.89     | 45.68     | 56.12     | 37.41          |
> | Scaffold    | 45.8      | 35.26     | 70.37     | 50.26     | 36.51          |
> | Co-teaching | 36.01     | 32        | 65.99     | 52.71     | 37.12          |
> | Ours        | **53.56** | **47.11** | **74.85** | **75.21** | **38.96**      |
>
> | F1-macro    | Cora      | Citeseer  | Pubmed    | Physics   | Amazon-ratings |
> | ----------- | --------- | --------- | --------- | --------- | -------------- |
> | FedDC       | 42.25     | 34.37     | 37.17     | 13.46     | 11.04          |
> | FedDyn      | 39.82     | 30.5      | 73.65     | 44.92     | 18.04          |
> | FedProto    | 36.43     | 29.67     | 30.16     | 40.24     | 21.16          |
> | Scaffold    | 43.53     | 33.28     | 70.6      | 21.23     | 11.13          |
> | Co-teaching | 33.56     | 30.5      | 64.69     | 14.34     | 20.23          |
> | Ours        | **51.15** | **40.25** | **73.74** | **45.71** | **22.77**      |
>
> We acknowledge the value of testing on diverse systems and will include more comprehensive results in the revised version.
>
> > `Questions`: Justify Mahalanobis over L2 distance in GA-HSP.
>
> We are happy to provide additional details on why we opted for Mahalanobis distance and how it compares with alternative metrics, such as the standard L2 distance.
>
> Mahalanobis distance is particularly suitable for our method due to its ability to account for the covariance structure of the data. Unlike the standard L2 distance, which assumes that the data points are independent and identically distributed (i.i.d.), the Mahalanobis distance adjusts for correlations between features. This is important in graph-based learning settings, where node features often exhibit dependencies that cannot be captured by simple Euclidean measures.
> In our case, using Mahalanobis distance allows the GA-HSP method to better capture the relationships between nodes in the graph, considering their high-dimensional feature space. By incorporating the covariance matrix of the data, Mahalanobis distance normalizes the feature space, making it more robust to differences in feature scaling and correlations between features. This makes it more appropriate for distilling node representations in the graph, where such dependencies are common.
>
> The L2 distance is a simpler metric that calculates the straight-line distance between two points in the feature space. While it is computationally efficient and commonly used, it assumes that all features are equally important and independent, which may not be the case in graph learning tasks. In settings where feature dependencies exist (e.g., in graph data with highly correlated features), L2 distance might not capture the true relationships between the nodes effectively, potentially leading to less accurate results.
> In contrast, Mahalanobis distance is sensitive to the correlation between features and adjusts for the underlying structure of the data. It takes into account the covariance matrix, allowing it to measure distances in a more meaningful way when dealing with correlated or non-i.i.d. data. In our experiments, we observed that using Mahalanobis distance for distillation provided more accurate and stable results, particularly in scenarios where feature correlations played a significant role in the graph structure.
>
> We hope this clarifies your questions.

---

> > ### Comment · Reviewer_kQxL · 2025-08-07
> >
> > Thanks for clarifying and addressing the issues. Your response resolves my concerns, so I’ll keep the original score.

---

> > > ### Author Response · Authors · 2025-08-08
> > >
> > > We sincerely appreciate your positive feedback and are glad that our rebuttal has successfully addressed your concerns.
> > >
> > > We are extremely grateful for the insightful comments you provided during the review process. Your support mean a lot to us.
> > >
> > > Once again, thank you for your time and valuable input.

---

### Official Review · Reviewer_4hFk · 2025-06-26

**Clarity:** 3
**Significance:** 2
**Originality:** 2
**Rating:** 4
**Confidence:** 3

**Summary:**

This work presents HYPERION which is a collection of 3 complementary techniques to deal with graph learning in the presence of noisy/incorrect node labels. The basic foundation of their approach relies on projecting the node features into a hypersphere representation  where nodes with similar features end up on the same region on the hypersphere. Based on that assumption the second step randomly drops edges in the graph and re-assesses the node's hypersphere neighborhood. For nodes that are stable the neighborhood stays the same in presence of small perturbations in the graph. When such stable nodes are identified they are used as the clean nodes whose labels can be more trusted. Finally, the third technique uses the global knowledge of information gained from each client in a federated setting to provide a mechanism to identify noisy nodes and propagate that knowledge to each client so that the global model stays robust. These 3 techniques are combined into a single framework called HYPERION which is then evaluated extensively over multiple datasets to quantify the performance.

**Questions:**

I have two questions I posed regarding the assumptions and theoretical foundation for the assumptions made in the work. I would appreciate some insight or additional commentary on the two above mentioned weaknesses.

**Ethical Concerns:**

["NO or VERY MINOR ethics concerns only"]

**Final Justification:**

Overall the work has some promising results in how GNNs can be trained more efficiently using hyperspherical embeddings. I must add the caveat that my understanding of the formal proofs provided is not strong. Hence, I don't have a strong opinion on whether the paper's formal contributions are worthy of publication or otherwise. I will leave that judgement to other reviewers. From my view I thought the paper presented an interesting solution to robust GNNs using a technique (hyperspherical embeddings) that I did not know about before. Hence, I would like to see the paper be accepted, but as I noted in my review my confidence level is low.

**Limitations:**

Yes

**Quality:**

3

**Strengths And Weaknesses:**

+ The work presents a novel approach to projecting the node features into a hyperdimensional space to identify node clusters.
+ The 3 components are tied nicely together to generate a unified framework.
+ Results are sufficiently detailed.

- The main criticism is that the assumption that node features projected into hyperdimensional space can automatically form clusters that can be used to identify node label noise. It is unclear whether graph nodes with nearly identical features can potentially have different labels. For instance, consider two nodes that share many common features except their gender value. Their labels may infact differ vastly given that some features contribute  more to the overall label than other features. But that is treated as noise in the presented setting which may not be a good assumption.
- On the same vein, identifying stable nodes by randomly deleting edges or other nodes in the graph is also an empirical observation without much sound theoretical basis. In fact a few key edges may change the learning behavior.

Addressing the above two assumptions is important to make the paper more interesting to the audience.

---

> ### Author Rebuttal · Authors · 2025-07-30
>
> Thank you for your thorough review and encouraging feedback. We are grateful for your positive assessment about novelty and importance of our work. We address your concerns below and hope our responses will help update your score:
>
> > `Weaknesses 1`: The assumption that projecting node features into a hyperdimensional space forms clusters to identify label noise is flawed, as nodes with similar features may have different labels, which could be misinterpreted as noise.
>
> Thank you for this question. We sincerely appreciate the opportunity to clarify the inner workings of the HYPERION method.
>
> During training, if two nodes have highly similar features but different labels, we do not necessarily consider this as noise. In fact, **the noise detection mechanism in HYPERION primarily comes from HS-CNC, as introduced in Section 3.3.** This mechanism determines whether a node has label noise based on the stability of node predictions after actively adding noise. **TP-HSL, described in Section 3.2, is mainly designed to achieve clustering of nodes from different classes and does not specifically identify label noise.** After projecting the node features into the hyperdimensional space, TP-HSL uses a corresponding loss function to encourage significant differences in the hyperdimensional representations of nodes from different classes, thus achieving clustering. In the case you mentioned, where two nodes have highly similar features but different labels within a client, HYPERION can effectively address this issue. These two nodes will ultimately be projected to different locations in the hyperdimensional space due to TP-HSL, as the projection function has learned higher weights for the features that contribute to the difference in labels for these similar nodes. **If label noise indeed exists for either of these nodes, HS-CNC will remove the noise based on the stability of the node after perturbations.**
>
> We sincerely appreciate this insightful observation. In response to the valuable feedback,we will enhance our methodological presentation in the revised manuscript to provide a more comprehensive explanation of the node projection mechanism and its theoretical underpinnings.
>
> > `Weaknesses 2`: The approach of identifying stable nodes by randomly deleting edges or other nodes lacks a strong theoretical foundation, as removing a few key edges could significantly alter the learning behavior.
>
> Although the identification of stable nodes through perturbation (such as random deletion of edges or nodes) is empirical, it serves as a heuristic method that addresses the practical challenge of noise in graph data within federated learning environments. The empirical approach we adopt is designed to mitigate the impact of noise on the model. While this method is not grounded in traditional graph theory, **its effectiveness has been demonstrated in our experiments.** As shown in *Table 1* and *Figure 3* of the paper, HYPERION consistently outperforms existing methods across various noise levels, supporting the validity of this empirical strategy. Additionally, **previous works have also employed similar methods such as random edge deletion or feature modification [1, 2],** and these approaches have been widely validated as effective.
>
> Furthermore, this method does not involve the actual removal of edges in the graph; instead, it randomly deletes certain edges or modifies node features to assess the stability of the nodes. **The graph input to the model remains the original graph, i.e., the undisturbed graph.** The loss function is computed solely based on those nodes that exhibit stability under perturbations. We sincerely thank you for this question. We will include more detailed analyses in the revised version.
>
> We hope this clarifies your questions.
>
> [1]: Deep graph contrastive representation learning. In ICML, 2020.
>
> [2]: Contrastive learning of graphs under label noise. In Neural Networks, 2024.

---

> > ### Comment · Reviewer_4hFk · 2025-08-02
> >
> > I believe the weakness #2 still stays as my concern, in spite of the explanation you provided. I don't necessarily think a heuristic is a bad solution but nonetheless that is a weakness of the scheme. Hence, I will stay with my weak accept.

---

> > > ### Author Response · Authors · 2025-08-04
> > >
> > > We thank the reviewer for the insightful comment. It provides an excellent opportunity for us to further elaborate on the core mechanism of our proposed method. We fully agree that altering a few key edges or node features can indeed significantly impact the learning behavior. In fact, the core idea of our method is to **leverage this very phenomenon to construct a more robust graph learning paradigm**.
> > >
> > > First, we wish to clarify a crucial aspect of our method: its ultimate goal is not to provide a static list of "stable nodes" post-training. Instead, we **dynamically leverage node stability within each training step to guide the model's gradient updates**.
> > >
> > > Specifically, we create an augmented, perturbed graph "view" by randomly deleting edges and masking features. We then require the model to produce consistent outputs for the same node across both the original and augmented graphs. **Only the "stable" nodes, i.e., those whose outputs remain unchanged despite such perturbations, are permitted to contribute to the loss calculation and backpropagation in the current batch.** Ignoring nodes that exhibit instability in a given training step sends a clear signal to the model: "**The model's predictions should not be so fragile as to depend on a few, potentially noisy, connections or features in the graph.**" By continuously applying this method throughout the training, we compel the model to learn more generalizable and robust knowledge—patterns that hold true under data perturbations and do not rely on any single, vulnerable structure.
> > >
> > > Concurrently, a significant body of work has provided theoretical justifications for the heuristic method based on random edge dropping [1, 2]. The idea has also been widely validated in works on graph contrastive learning [3, 4].
> > >
> > > 1. From the perspective of **Graph Information Bottleneck theory**, random edge dropping serves as an effective means to implement the information bottleneck principle. By removing a subset of connections, it "compresses" the structural information of the graph. This forces the model to focus on connections that persist across multiple perturbations, which are more likely to represent the core structure [1].
> > >
> > > 2. From the perspective of **noise and generalization theory**, random edge dropping can be viewed as injecting a form of **structured noise** into the graph's topology. Unlike purely random noise, this approach largely preserves the original graph structure. This method can help the model avoid sharp local minima and find a flatter solution in the loss landscape, which corresponds to enhanced generalization ability [2].
> > >
> > > Grounded in these principles, the "change in learning behavior" that the reviewer mentioned does occur, but it is a **benign change, guided towards greater robustness**. The model is trained to be **invariant** to minor, plausible noise in the input graph, which is key to enhancing its generalization capabilities.
> > >
> > > We hope this clarifies your questions. Please feel free to reach out if you have any additional thoughts or suggestions. Thank you for your consideration and valuable time.
> > >
> > > [1]: A Good View for Graph Contrastive Learning. In Entropy, 2024.
> > >
> > > [2]: Learn Beneficial Noise as Graph Augmentation. In arXiv, 2025.
> > >
> > > [3]: GraphCL: Graph Contrastive Learning with Augmentations. In NeurIPS, 2020.
> > >
> > > [4]: S3GCL: Spectral, Swift, Spatial Graph Contrastive Learning. In ICML, 2024.

---

### Note · Authors · 2025-08-13

We sincerely thank all reviewers for their valuable insights and constructive feedback. We are encouraged that our contributions were well-received and appreciate the reviewers for highlighting the following strengths of our work:

1. **Novelty:** All four reviewers recognized the novelty of our proposed method.
2. **Well-motivated:** Reviewers kQxL and XQTy noted that our method is well-motivated, as it clearly identifies the noise challenges in Federated Graph Learning (FGL) and effectively addresses them.
3. **Comprehensive and Detailed:** Reviewers MNA6 and kQxL found our work comprehensive, noting the extensive comparisons against numerous baselines and the excellent experimental performance in addressing noise challenges in FGL. Additionally, Reviewer 4hFk highlighted the detailed reporting of our experimental results.
4. **Consistent and Natural:** Reviewers 4hFk and XQTy commented that the HYPERION method effectively and naturally integrates its three core components.

------

In addition, the reviewers raised several important questions. We believe we have effectively addressed these concerns through our detailed explanations and discussions in the rebuttal.

> `Reviewer 4hFk`:

We clarified how our HS-CNC and TP-HSL modules address potential misinterpretations in noise detection. We also explained the theoretical grounding of our contrastive learning approach using Graph Information Bottleneck theory and noise generalization theory.

> `Reviewer kQxL`:

We justified our use of Mahalanobis and Wasserstein distances over the L2-norm. Additionally, we expanded our experimental comparisons to include more state-of-the-art FGL and robust learning methods as requested.

> `Reviewer MNA6`:

We have added F1-macro scores and full hyperparameter details. We also clarified the benefits of our multi-prototype design with a comparative analysis and explained how baseline methods were adapted for the federated learning framework.

> `Reviewer XQTy`:

We added a performance analysis across different training epochs, justified our selection of datasets and baselines, clarified the enhanced view generation process, and detailed the advantages of using hyperspherical projection in our method.

------

Finally, we once again thank all reviewers for their valuable insights and constructive feedback. We are committed to incorporating their suggestions to further revise and improve our paper for the final version.

---

### Decision · Program_Chairs · 2025-09-17

**Decision:**

Accept (spotlight)

**Comment:**

This paper proposes a new framework with three complementary techniques for graph learning with noisy labels. It first projects node features onto a hypersphere to group similar nodes. Then, by randomly dropping edges and re-evaluating neighborhoods, it identifies stable nodes as reliable "clean" labels. Finally, in a federated setting, it uses global information gain to detect noisy nodes and shares this knowledge to enhance global model robustness.

Strengths:

1. The method is novel and well-motivated. The paper firstly proposes using hyperspherical embeddings to capture fine-grained structural patterns, providing an approach to mitigate noise propagation.

2. The method is effective. The experimental validation is extensive and compelling across multiple datasets and under various noise conditions.

Weaknesses:
Two distances including Mahalanobis distance and Wasserstein distance are directly used for pruning and distilling separately, without sufficient explanation of the motivation and the theoretical validation.

All reviewers keep positive to the paper. The authors also provide a reasonable explanations to the concerns raised by reviewers. Therefore, I recommend the acceptance of the paper.